# The neural origin for asymmetric coding of surface color in the primate visual cortex

Yujie Wu [1,2,6], Minghui Zhao [3,4,6], Haoyun Deng[3,4], Tian Wang[1,5], Yumeng Xin[3,4], Weifeng Dai[1], Jiancao Huang[1], Tingting Zhou[1], Xiaowen Sun[1], Ning Liu [3,4] ✉ & Dajun Xing [1] ✉

The coding privilege of end-spectral hues (red and blue) in the early visual cortex has been reported in primates. However, the origin of such bias remains unclear. Here, we provide a complete picture of the end-spectral bias in visual system by measuring fMRI signals and spiking activities in macaques. The correlated end-spectral biases between the LGN and V1 suggest a subcortical source for asymmetric coding. Along the ventral pathway from V1 to V4, red bias against green peaked in V1 and then declined, whereas blue bias against yellow showed an increasing trend. The feedforward and recurrent modifications of end-spectral bias were further revealed by dynamic causal modeling analysis. Moreover, we found that the strongest end-spectral bias in V1 was in layer 4C$\beta$. Our results suggest that end-spectral bias already exists in the LGN and is transmitted to V1 mainly through the parvocellular pathway, then embellished by cortical processing.

The spectral reflectance of a surface provides indispensable color information for primates and humans during natural vision. Surface color helps us find things quicker and recognize/remember them better[1] by improving edge classification[2], providing cues for object segregation[3,4], and fastening visual encoding and retrieval of imaged information from memory[5]. Investigating how the visual cortex encodes color is essential for understanding the formation of category effects on color perception.

Rather than uniform representation of spectral information, the primary visual cortex (V1) exhibits asymmetric coding of different wavelengths of light: a strong bias on end-spectral colors (e.g., red and blue) relative to mid-spectral colors (green and yellow)[6–8]. Color opponency (red–green, blue–yellow) arises as early as the horizontal cells of the retina[9] and in ganglion cells in macaques[10,11], and it is transmitted by parvocellular (red–green opponent) and koniocellular neurons (blue–yellow opponent) in the lateral geniculate nucleus (LGN)[12–14]. In V1, end-spectral colors evoked stronger responses[15,16], involved more neurons in the encoding process[17], and induced

stronger gamma oscillations (~ 30–80 Hz) in both monkey[18,19] and human visual cortex[20]. Although end-spectral bias has been found through various techniques in humans and monkeys, its neural origin is still unclear.

Two potential mechanisms for generating spectral bias need to be tested: the cortical mechanism and the precortical feedforward mechanism. The cortical hypothesis assumes that the color response bias may emerge within V1 from the input layer to the output layer due to the circuits in the output layers[18,21]. Based on studies in the luminance domain, the cortical mechanism in the V1 output layer contributes to black dominance[22,23]. Compared to achromatic stimuli, the cortical mechanism evoked by chromatic stimuli can be pathway-dependent[24]. More specifically, considering that there is a larger segregated clustering of neurons preferring red and blue than green and yellow[25,26], the horizontal connection strength in the output layer may be unbalanced for different colors, which may lead to the end-spectral bias in V1. On the other hand, according to the precortical feedforward hypothesis, the response bias in V1 may be inherited from the LGN[8], the

[1]State Key Laboratory of Cognitive Neuroscience and Learning & IDG/McGovern Institute for Brain Research, Beijing Normal University, Beijing 100875, China. [2]Princeton Neuroscience Institute, Princeton University, Princeton, NJ 08544, USA. [3]State Key Laboratory of Brain and Cognitive Science, Institute of Biophysics, Chinese Academy of Sciences, Beijing 100101, China. [4]College of Life Sciences, University of Chinese Academy of Sciences, Beijing 100049, China. [5]College of Life Sciences, Beijing Normal University, Beijing 100875, China. [6]These authors contributed equally: Yujie Wu, Minghui Zhao. ✉e-mail: liuning@ibp.ac.cn; dajun_xing@bnu.edu.cn

main relay station of signals leaving the retina to the cortex. As found in black–white asymmetry/polarity bias[27,28], precortical nonlinear processing in separate pathways[29] or unbalanced projecting strength[30,31] may also account for the end-spectral bias in V1. Whether the color domain shares a similar feedforward mechanism still lacks direct evidence.

In this study, by using electrophysiology data and fMRI results recorded in macaques, we aim to reveal the mechanisms generating end-spectral bias and the cortical mechanisms modulating it. The combination of intracortical laminar recording and large-scale brain imaging in the same species can help us simultaneously uncover color processing in separate visual pathways and at multiple stages. We first compared the electrophysiological findings with fMRI results to confirm the consistency of end-spectral bias in V1 for the two measurements. Then, we investigated the end-spectral bias in the LGN, V1, V2 and V4 along the ventral visual pathway. To further explore the contribution of feedforward drive and cortical modulation, we employed stochastic dynamic causal modeling (DCM). We finally used linear probes to simultaneously record neurons from all layers in V1 to test the pathway-specific nature in the feedforward stream of end-spectral bias.

## Results

In the electrophysiology experiments, we recorded the multiunit activity (MUA) and local field potential (LFP) in V1 by Utah arrays in two monkeys (DQ and DP) and by a linear probe in another two monkeys (DS and QQ) while they performed a fixation task (Fig. 1a). During each trial, the four monkeys were trained to fixate around a small dot in the center of the screen, while a square (4°) with one of four hues, all equiluminant to the gray background (see cone contrast for each visual stimulus in Supplementary Table 1), or a square of black or white with the same absolute contrast value (Supplementary Table 1) was presented for 2 s. The receptive fields (RFs) were estimated by a two-dimensional Gaussian function based on the response map obtained in the sparse noise experiment (Supplementary Fig. 1). In each trial, the center or edge of the square was shown in the RFs of the recorded sites (Fig. 1a). In the fMRI experiment, responses from the LGN and cortical visual areas driven by similar squares were recorded in another four monkeys. To more efficiently evoke responses in the bilateral visual pathway, we simultaneously presented two uniform squares (4°) to both hemispheres (Fig. 1e) for 2 s during the stimulus block.

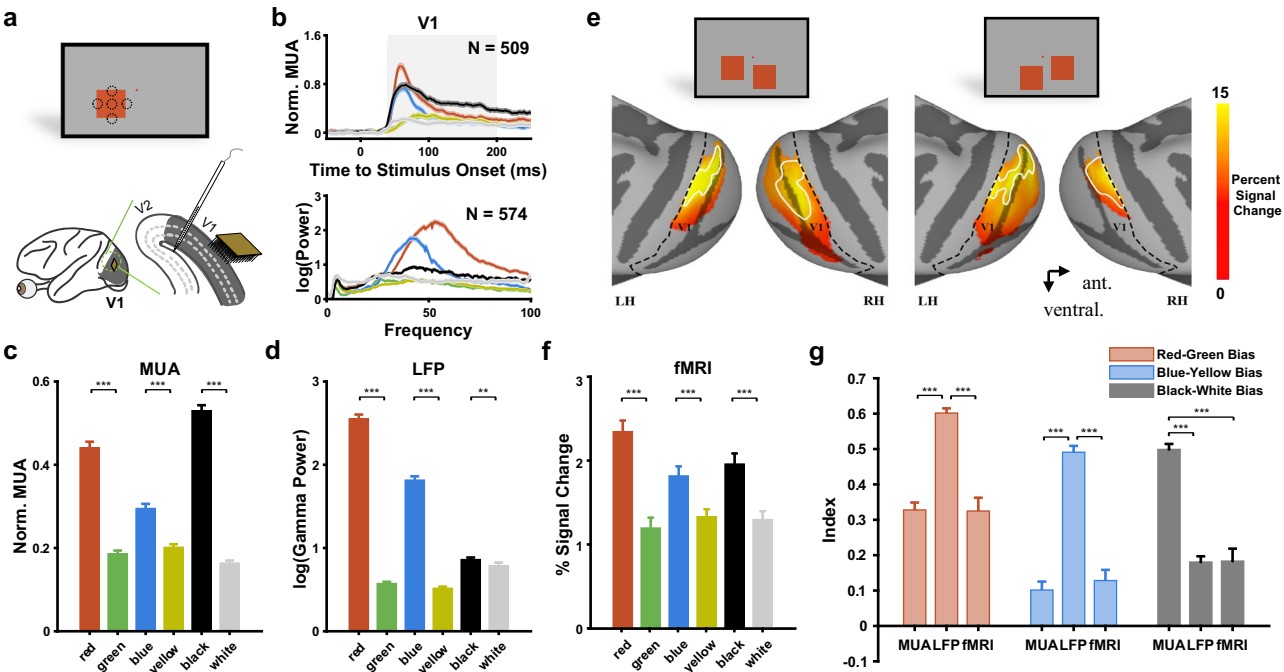

**Fig. 1 | Biased color and luminance coding from physiological and imaging data. a** An example of stimuli used in the electrophysiological experiment. A Utah array (Blackrock, 10 × 10, length 1 mm, interchannel spacing 400 μm) was implanted in V1 of two monkeys (DQ and DP). A linear probe (Plexon, 24 channels, interchannel spacing 100 μm) was perpendicularly placed into the V1 of the monkey DS or QQ on a daily basis. During each trial of the fixation task, the center or edge of a uniform square (red, green, blue, yellow, black, white) was shown in the RFs of the recorded sites to match the fMRI responses to the whole square stimulus (Supplementary Fig. 1). Here, we show a simplified illustration of 5 RF locations relative to the stimulus outline. **b** The spiking activity of multiple units and power spectrum from local field potentials for each type of stimulus are shown in the corresponding color. **c** Normalized MUA during 40–200 ms after stimulus onset. ***$p < 0.001$ in post hoc comparisons of one-way ANOVA (two-sided, $n = 509$ channels, red vs green, $p = 4.58 \times 10^{-66}$, blue vs yellow, $p = 1.3 \times 10^{-9}$, black vs white, $p = 1.38 \times 10^{-129}$). **d** The power spectrum was fitted by a linear model composed of Gaussian and exponential decay. Gamma power was estimated by the fitted power relative to the fitted blank power in the peak location (frequency) of the Gaussian. **p < 0.01, ***p < 0.001 in post hoc comparisons following one-way ANOVA (two-

sided, $n = 574$ channels, red vs green, $p = 0$, blue vs yellow, $p = 3.5 \times 10^{-149}$, black vs white, $p = 0.008$). **e** Examples of two stimulus configurations used in the fMRI experiment are shown in the top panels of (**e**). Activation maps (averaged percent signal changes across 6 types of stimuli thresholded at t > 6) and V1 ROI (encircled by white lines) in monkey Q are shown on lateral views of the inflated cortex of the macaque template [(the bottom panel of (**e**)]. The borders of V1 are indicated by the black dashed lines. **f** Averaged fMRI responses to three color pairs across all four subjects in V1. ***$p < 0.001$ in comparisons of the generalized linear mixed model (GLMM) analyses (two-sided, $n = 102$ runs, red vs green, $p = 1.15 \times 10^{-13}$, blue vs yellow, $p = 1.07 \times 10^{-5}$, black vs white, $p = 1.79 \times 10^{-7}$). **g** Bias indices detected by three approaches (n of MUA = 509 channels, n of LFP = 574 channels, n of fMRI = 102 runs). ***$p < 0.001$ in Bonferroni-corrected post hoc comparisons following one-way ANOVA (two-sided, red–green bias: MUA vs LFP, $p = 5.78 \times 10^{-28}$, LFP vs fMRI, $p = 3.43 \times 10^{-10}$; blue–yellow bias: MUA vs LFP, $p = 7.1 \times 10^{-39}$, LFP vs fMRI, $p = 3.51 \times 10^{-12}$; black–white bias, MUA vs LFP, $p = 4.96 \times 10^{-42}$, MUA vs fMRI, $p = 1.69 \times 10^{-14}$). Data are presented as mean + SE. Source data are provided as a Source Data file.

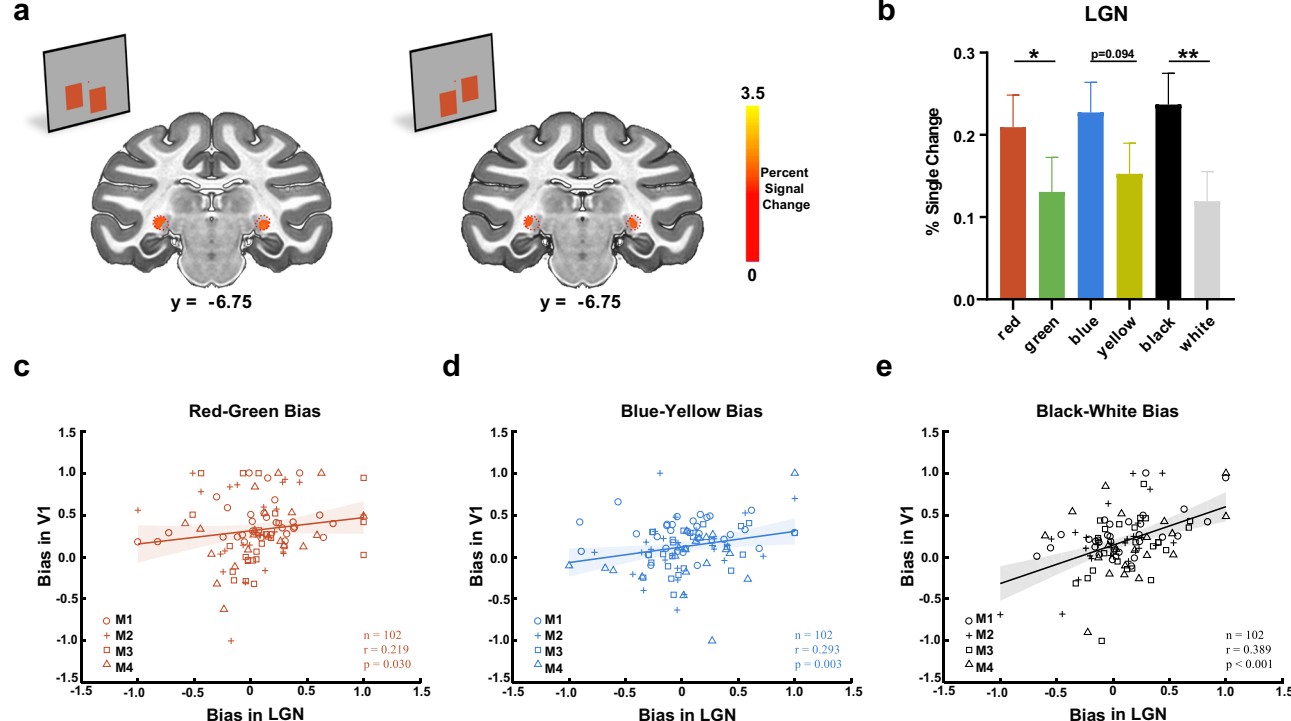

**Fig. 2 | End-spectral and polarity bias in the LGN and its relationship with V1.**
**a** Activation maps (averaged percent signal changes across 6 types of stimuli thresholded at t > 1.96) in the LGN of monkey Q are shown on coronal slices of the macaque template. The borders of the LGN are indicated by the red dotted lines. **b** Averaged fMRI responses to three color pairs across all four subjects in the LGN. *$p < 0.05$, **$p < 0.01$, GLMM analyses (two-sided, red vs green, $p = 0.039$, blue vs yellow, $p = 0.094$, black vs white, $p = 0.008$). Data are presented as mean + SE

($n = 102$ runs). **c**–**e** Correlations between LGN and V1 for the red–green bias, blue–yellow bias, and black–white bias (two-tailed Spearman correlation, $n = 102$ runs, red–green bias, $p = 0.030$, blue–yellow bias, $p = 0.003$, black–white bias, $p = 7.51 \times 10^{-5}$). Each monkey is represented by a different symbol. The shaded region represents 95% confidence intervals. Source data are provided as a Source Data file.

## Consistent end-spectral bias in electrophysiology and imaging data

With electrophysiology recordings (Fig. 1b), we found end-spectral color dominance in V1 for both spiking activity (Fig. 1c, $F_{5,3048} = 221.83$, $MS_e = 0.05$, $p < 0.001$) and gamma oscillation (Fig. 1d, $F_{5,3438} = 599.55$, $MS_e = 0.64$, $p < 0.001$): MUA and gamma power of red (mean ± s.e., MUA: $0.44 \pm 0.01$, gamma: $2.55 \pm 0.05$) and blue (MUA: $0.30 \pm 0.01$, gamma: $1.82 \pm 0.04$) were stronger than their opposite colors, green (MUA: $0.18 \pm 0.01$, gamma: $0.58 \pm 0.02$) and yellow (MUA: $0.20 \pm 0.01$, gamma: $0.52 \pm 0.02$), respectively (post hoc comparisons, $ps < 0.001$). We also found significant black dominance in spiking activity (Fig. 1c): the black response ($0.53 \pm 0.01$) was not only stronger than the white response ($0.16 \pm 0.01$) but also stronger than all color responses (post hoc comparisons, $ps < 0.001$).

Similar to what was found in the MUA and LFP, the results based on fMRI responses (Fig. 1e, see the unthresholded map in Supplementary Fig. 2) showed consistent end-spectral and polarity bias patterns in V1 (Fig. 1f, red vs. green: $t(202) = 7.97$, $p < 0.001$; blue vs. yellow: $t(202) = 4.52$, $p < 0.001$; black vs. white: $t(202) = 5.41$, $p < 0.001$, GLMM analysis): the red ($2.35 \pm 0.13$), blue ($1.82 \pm 0.11$), black ($1.96 \pm 0.12$) stimuli evoked stronger responses than their opposite color/polarity, green ($1.20 \pm 0.12$), yellow ($1.33 \pm 0.09$) and white ($1.30 \pm 0.10$), respectively. These consistent findings indicated that end-spectral and polarity bias existed in V1 and could be reliably assessed by the fMRI method.

To further characterize and compare the spectral bias from physiological and imaging data, we defined bias indices for the three pair colors/luminance (red vs. green, blue vs. yellow and black vs. white). We found comparable red–green bias and blue–yellow bias between

MUA (red bias: $0.33 \pm 0.02$, blue bias: $0.10 \pm 0.02$) and fMRI (red bias: $0.32 \pm 0.04$, blue bias: $0.13 \pm 0.03$) responses (Fig. 1g, ps = 1 in Bonferroni-corrected post hoc comparisons following one-way ANOVA; red–green bias: $F_{2,1182} = 70.99$, $MS_e = 0.16$, $p < 10^{-10}$, blue–yellow bias: $F_{2,1182} = 99.59$, $MS_e = 0.22$, $p < 10^{-10}$). For luminance, black–white bias in the MUA ($0.33 \pm 0.02$) was stronger than that in both the LFP ($0.18 \pm 0.02$, $p < 10^{-10}$) and fMRI ($0.18 \pm 0.04$, $p < 10^{-10}$ in a Bonferroni-corrected comparison following one-way ANOVA, $F_{2,1182} = 108.63$, $MS_e = 0.14$, $p < 10^{-10}$). The similar spectral bias between electrical signals and fMRI signals in the macaque V1 ensures that we can take advantage of the fMRI method to examine and compare different regions (including the LGN, V1, V2 and V4) in the visual pathway. Thus, we will be able to address two important questions regarding end-spectral bias: 1) What are the generation mechanisms for the biases on end-spectral hues? and 2) How do the biases on end-spectral hues change along the visual pathway?

## End-spectral and polarity bias in LGN

After confirming a similar strength of spectral bias between the electrical signal and fMRI signal in macaque V1, we tested the two mechanisms for such bias. If the end-spectral bias in V1 was inherited from LGN, then we should find comparable bias in LGN, which should be correlated with bias in V1. Otherwise, if the end-spectral bias emerged through a cortical mechanism, then we may find different biases in the LGN. Therefore, we took advantage of fMRI to examine the LGN (Fig. 2a). We found that LGN was also activated more by red ($0.21 \pm 0.04$), blue ($0.23 \pm 0.04$), and black ($0.24 \pm 0.04$) than their corresponding paired color/polarity [i.e., green ($0.13 \pm 0.04$), yellow

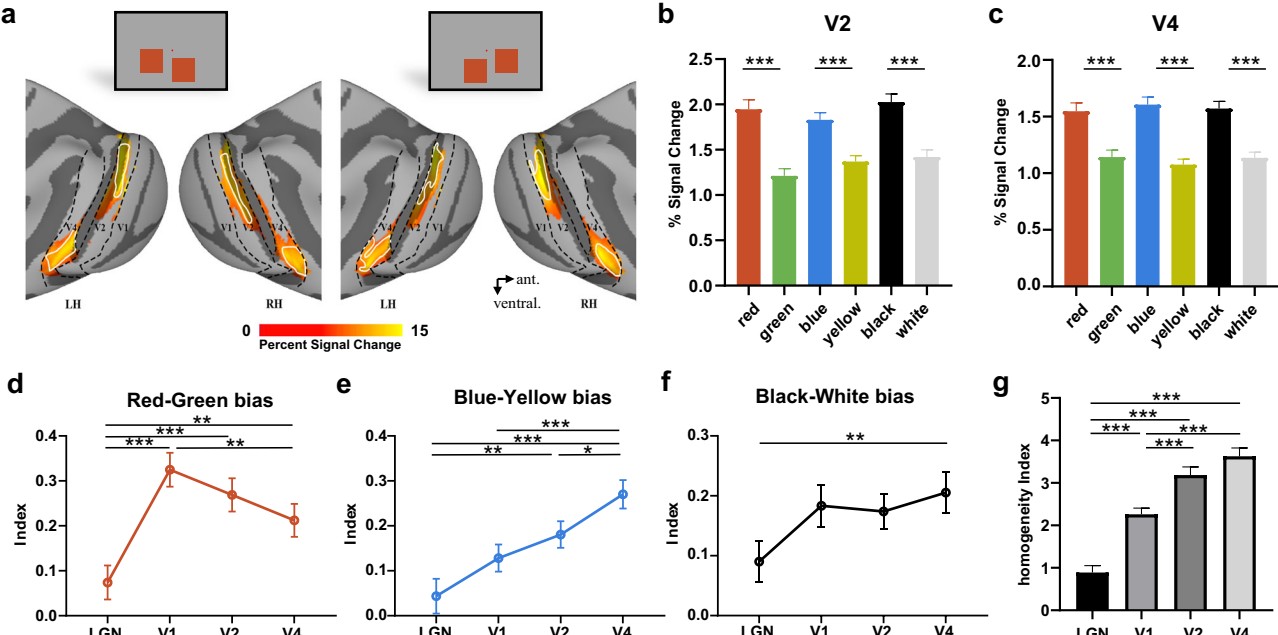

**Fig. 3 | End-spectral and polarity bias in V2 and V4. a** Activation maps (averaged percent signal changes across 6 types of stimuli thresholded at t > 10) and ROIs (encircled by white lines) in V2 and V4 of monkey Q are shown on lateral views of the inflated cortex of the macaque template. The borders of V1, V2 and V4 are indicated by the black dashed lines. **b, c** Averaged fMRI responses to three color pairs across all four subjects in V2 and V4, respectively. *$p < 0.05$, GLMM analysis (two-sided, V2: red vs green, $p = 2.45 \times 10^{-13}$, blue vs yellow, $p = 2.21 \times 10^{-10}$, black vs white, $p = 7.33 \times 10^{-15}$; V4: red vs green, $p = 9.70 \times 10^{-10}$, blue vs yellow, $p = 0$, black vs white, $p = 2.22 \times 10^{-13}$). **d–f** The change trends of red–green bias, blue–yellow bias, and black–white bias from LGN to V4. *$p < 0.05$, **$p < 0.01$, ***$p < 0.001$, post hoc

comparisons of GLMM with Bonferroni corrections (two-sided, red–green bias: LGN vs. V1, $p = 2.33 \times 10^{-7}$, LGN vs. V2, $p = 4.49 \times 10^{-5}$, LGN vs. V4, $p = 0.005$, V1 vs. V4, $p = 0.004$; blue–yellow bias: LGN vs. V2, $p = 0.002$, LGN vs. V4, $p = 1.53 \times 10^{-7}$, V1 vs. V4, $p = 1.36 \times 10^{-4}$, V2 vs. V4, $p = 0.012$; black–white bias: LGN vs. V4, $p = 0.006$). **g** Color homogeneity across visual areas. ***$p < 0.001$, post hoc comparisons of GLMM with Bonferroni corrections (two-sided, LGN vs. V1, $p = 1.94 \times 10^{-11}$, LGN vs. V2, $p = 0$, LGN vs. V4, $p = 0$, V1 vs. V2, $p = 2.25 \times 10^{-6}$, V1 vs. V4, $p = 2.32 \times 10^{-10}$). Data are presented as mean + SE (**b, c, g**) or +/- SE (**d, e, f**) ($n = 102$ runs). Source data are provided as a Source Data file.

(0.15 ± 0.04), and white (0.12 ± 0.04), respectively] (Fig. 2b: red vs. green: $t(202) = 2.07$, $p = 0.039$; blue vs. yellow: $t(202) = 1.68$, $p = 0.094$; black vs. white: $t(202) = 2.69$, $p = 0.008$, GLMM analysis). These results indicate that the end-spectral bias starts from the precortical stage.

To investigate whether the end-spectral and polarity bias in the LGN is related to the bias in V1, we calculated the end-spectral and polarity bias index for the LGN and V1 for each run and conducted Spearman correlation to assess the relationship between the two brain regions. We controlled for individual monkeys as a variable by converting their identifiers into binary values. Significant correlations were found for all three pairs (Fig. 2c–e), further confirming that the end-spectral and polarity bias of V1 may arise from LGN. Notably, even after we removed the bias pairs with extreme values (absolute value surpassed the threshold: 0.75), the correlations between LGN and V1 remained significant (Supplementary Fig. 3). Furthermore, we found that the red bias in V1 was significantly higher than the red bias in LGN ($t(404) = 5.60$, $p < 0.001$, post hoc comparisons of GLMM with Bonferroni corrections), while the blue and black bias was comparable in the two areas (blue–yellow bias: $t(404) = 2.05$, $p = 0.081$; black–white bias: $t(404) = 2.47$, $p = 0.070$, post hoc comparisons of GLMM with Bonferroni corrections) (Fig. 3d–f). Overall, the existence and covariance of end-spectral bias in the LGN suggest a feedforward precortical mechanism for the generation of the phenomenon in V1. The increasing red bias from the LGN to V1 suggests that the red bias may be magnified during the projection from the LGN to V1 or magnified through the feedback connection in cortical areas. The downstream cortical processing of hues may further lead to the change in end-spectral bias from V1 to V4. In the next two parts, we will illustrate such a transition along the visual pathway and explore the possible

subcortical-cortical circuit dynamics contributing to the generation and modification of end-spectral bias.

**Transition of end-spectral and polarity bias in the ventral visual pathway**

After identifying the feedforward drive of end-spectral bias in the LGN, we aimed to explore the cortical modification of the response asymmetry to color pairs in V2 and V4 (Fig. 3a, see the unthresholded map in Supplementary Fig. 2). First, we found that both V2 and V4 responded more strongly to red (V2: 1.95 ± 0.10, V4: 1.55 ± 0.07), blue (V2: 1.83 ± 0.08, V4: 1.61 ± 0.07), and black (V2: 2.03 ± 0.09, V4: 1.57 ± 0.06) than to their opposite color/polarity, green (V2: 1.21 ± 0.08, V4: 1.14 ± 0.06), yellow (V2: 1.37 ± 0.06, V4: 1.08 ± 0.05), and white (V2: 1.42 ± 0.08, V4: 1.34 ± 0.05), respectively (Fig. 3b, c, V2: red vs. green: $t = 7.85$, $p < 0.001$; blue vs. yellow: $t = 6.69$, $p < 0.001$; black vs. white: $t = 8.41$, $p < 0.001$; V4: red vs. green: $t = 6.42$, $p < 0.001$; blue vs. yellow: $t = 9.41$, $p < 0.001$; black vs. white: $t = 7.86$, $p < 0.001$, GLMM analysis). Moreover, the biases in V2 and V4 were significantly correlated with the biases in V1 and LGN for all three pairs (Table 1), indicating the existence and inheritance of end-spectral bias in the downstream visual areas from LGN and V1. Note that the correlation between two regions might be mediated by a third region. For instance, V1 and V2 might act as mediators in the observed correlation between LGN and V4 (Supplementary Table 2), which could complicate the interpretation of the correlation and its strength. To provide a more comprehensive understanding of the underlying neural circuits, we conducted dynamic causal modeling (DCM) and laminar activity analyses after the fact.

Second, we focused on illustrating the transition of end-spectral bias along the visual pathway. We found different trends in end-spectral and polarity bias from V1 to V2/V4 for different color pairs

**Table 1 | Correlations of bias indices among the LGN, V1, V2 and V4**

| | Correlation Spearman r | V1 r(p) | V2 r(p) | V4 r(p) |
|---|---|---|---|---|
| Red–Green bias | LGN | 0.219 ($p = 0.030$)* | 0.313 ($p = 0.002$)** | 0.330 ($p < 0.001$)*** |
| | V1 | | 0.580 ($p < 0.001$)*** | 0.514 ($p < 0.001$)*** |
| | V2 | | | 0.652 ($p < 0.001$)*** |
| Blue–Yellow bias | LGN | 0.293 ($p = 0.003$)** | 0.348 ($p < 0.001$)*** | 0.439 ($p < 0.001$)*** |
| | V1 | | 0.436 ($p < 0.001$)*** | 0.489 ($p < 0.001$)*** |
| | V2 | | | 0.503 ($p < 0.001$)*** |
| Black–White bias | LGN | 0.389 ($p < 0.001$)*** | 0.359 ($p < 0.001$)*** | 0.509 ($p < 0.001$)*** |
| | V1 | | 0.611 ($p < 0.001$)*** | 0.544 ($p < 0.001$)*** |
| | V2 | | | 0.612 ($p < 0.001$)*** |

*$p < 0.05$, **$p < 0.01$, ***$p < 0.001$, two-sided Spearman correlation.

(Fig. 3d–f). The red–green bias started to decrease from V1 to V4 (V1 vs. V4: $t(404) = 3.22$, $p = 0.005$; V1 vs. V2: $t(404) = 1.56$, $p = 0.177$; V2 vs. V4: $t(404) = 1.71$, $p = 0.177$, post hoc comparisons of GLMM with Bonferroni corrections). In contrast, the blue–yellow bias showed a gradually increasing trend from V1 to V4 (V1 vs. V4: $t(404) = -4.24$, $p < 0.001$; V1 vs. V2: $t(404) = -1.60$, $p = 0.110$; V2 vs. V4: $t(404) = -2.90$, $p = 0.012$, post hoc comparisons of GLMM with Bonferroni corrections). At the same time, black–white bias was relatively stable from V1 to V4 (V1 vs. V4: $t(404) = -0.68$, $p = 0.995$; V1 vs. V2: $t(404) = 0.30$, $p = 0.995$; V2 vs. V4: $t(404) = -1.13$, $p = 0.781$, post hoc comparisons of GLMM with Bonferroni corrections).

A previous study assumed a more uniform representation for different hues from V1 to V4 by showing more dissociable and more equal hue clusters in spatial organization[17]. Here, we qualitatively investigated the representation uniformity on neural response across all four basic colors from LGN to V4 by defining the homogeneity index for each area as the reciprocal of the standard deviation multiplied by the mean value of color responses (Fig. 3g). We found a gradual increase in the color homogeneity index from the LGN to V4 (LGN vs. V1: $t = -7.12$, $p < 0.001$; LGN vs. V2: $t = -10.57$, $p < 0.001$; LGN vs. V4: $t = -11.78$, $p < 0.001$; V1 vs. V2: $t = -4.94$, $p < 0.001$; V1 vs. V4: $t = -6.68$, $p < 0.001$; V2 vs. V4: $t = -1.94$, $p = 0.053$, post hoc comparisons of GLMM with Bonferroni corrections), suggesting a rise in uniformity of color representation. Note that the lower homogeneity index in the LGN might be attributed to the greater variability (Supplementary Fig. 4a). As such, we mainly focused on the homogeneity changes from V1 to V4. Similar results were obtained when we treated the contrast-to-noise ratio (CNR) as a covariate and reanalyzed the data (Supplementary Fig. 4b). As shown in Supplementary Fig. 5, responses from V1 to V4 gradually converged across the four colors, with red and blue exhibiting closer responses and yellow and green converging as well. Moreover, the reduction in red–green bias might also contribute to the uniformity of color representation V1 to V4 (Fig. 3d).

In this study, individual images were aligned to the symmetric NIMH Macaque Template (NMT) v2. The areal boundaries were established based on the D99 atlas in NMT v2 space rather than individual receptive field maps. Consequently, the precise accuracy of the areal boundaries might be subject to variation. Moreover, one of the stimuli was presented close to the vertical meridian, representing the boundary of cortical areas. As such, disentangling the area to which the responses correspond was a challenge due to the proximity to this boundary. To evaluate the potential impact of these limitations, we employed two additional approaches to ROI definitions (see details in Supplementary Note 1, see also supplementary Figs. 6–8). Both of them yielded consistent results with those shown in the main results. Therefore, it is plausible that our findings were minimally influenced by the above-mentioned limitations.

## Dynamic causal modeling of end-spectral and polarity bias from LGN to V4

The abovementioned analyses indicated that the end-spectral bias already exists in the LGN and changes in the downstream cortical areas. To further investigate whether the end-spectral and polarity bias of V1 was magnified through feedforward connections from the LGN or through feedback connections from V2/V4, we conducted DCM analyses and examined the effective connectivity among the LGN, V1, V2, and V4 during color and luminance stimulus presentation. Connection parameters with strong evidence (i.e., posterior probability > 0.95) were kept in the full model (solid lines in Fig. 4a), and two feedback parameters from V2 to V1 and from V1 to LGN were not significant. We found that all other connections except those between V2 and V4 were significantly positive, suggesting that there was robust feedforward connectivity from the LGN to V1, V1 to V2 and V1 to V4. Therefore, V1 received both a significant feedforward drive from the LGN and a feedback drive from V4 during the uniform color/luminance presentation.

Next, to compare the feedforward/feedback connectivity (e.g., from LGN to V1, V1 to V2 and V1 to V4) between the dominant color/polarity and the opponent color/polarity, we calculated the differences in the modulatory connections caused by the two colors in each color pair for the significant connections identified in Matrix A. Contrasts that exceeded the threshold ($p < 0.05$) are shown as solid lines in Fig. 4b–d. We found that red, blue, and black stimuli had stronger feedforward connections than green, blue, and white stimuli, respectively (Fig. 4e–g: red vs. green: LGN-V1: $t = 4.09$, $p < 0.001$; V1-V2: $t = 5.41$, $p < 0.001$; V1-V4: $t = 4.72$, $p < 0.001$; blue vs. yellow: LGN-V1: $t = 3.05$, $p = 0.003$; V1-V2: $t = 4.25$, $p < 0.001$; V1-V4: $t = 5.41$, $p < 0.001$; black vs. white: LGN-V1: $t = 3.07$, $p = 0.002$; V1-V2: $t = 5.74$, $p < 0.001$; V1-V4: $t = 6.38$, $p < 0.001$). These results demonstrated that functional connectivity from the LGN to V1 and then V1 to V2/V4 were more modulated by red, blue, and black than their paired colors (green, yellow, and white, respectively). Notably, we also looked into the modulatory effects evoked by stimuli on the feedback connections (i.e., V1-LGN and V2-V1), which were not significant in Matrix A (see Supplementary Fig. 9 for the results of all the connections). The differences in modulatory effects between red and green, blue and yellow, and black and white on feedback connections were minor or negative. Note that the negative difference indicated a stronger inhibitory feedback modulation effect evoked by red, blue, and black stimuli than by green, yellow, and white stimuli, respectively. In addition, red and black modulated the self-connections of the LGN and V1 more than green and white, indicating that the LGN and V1 were more sensitive to red and black than to green and white. Taken together, these results suggested a feedforward mechanism for the magnification of end-spectral bias in V1.

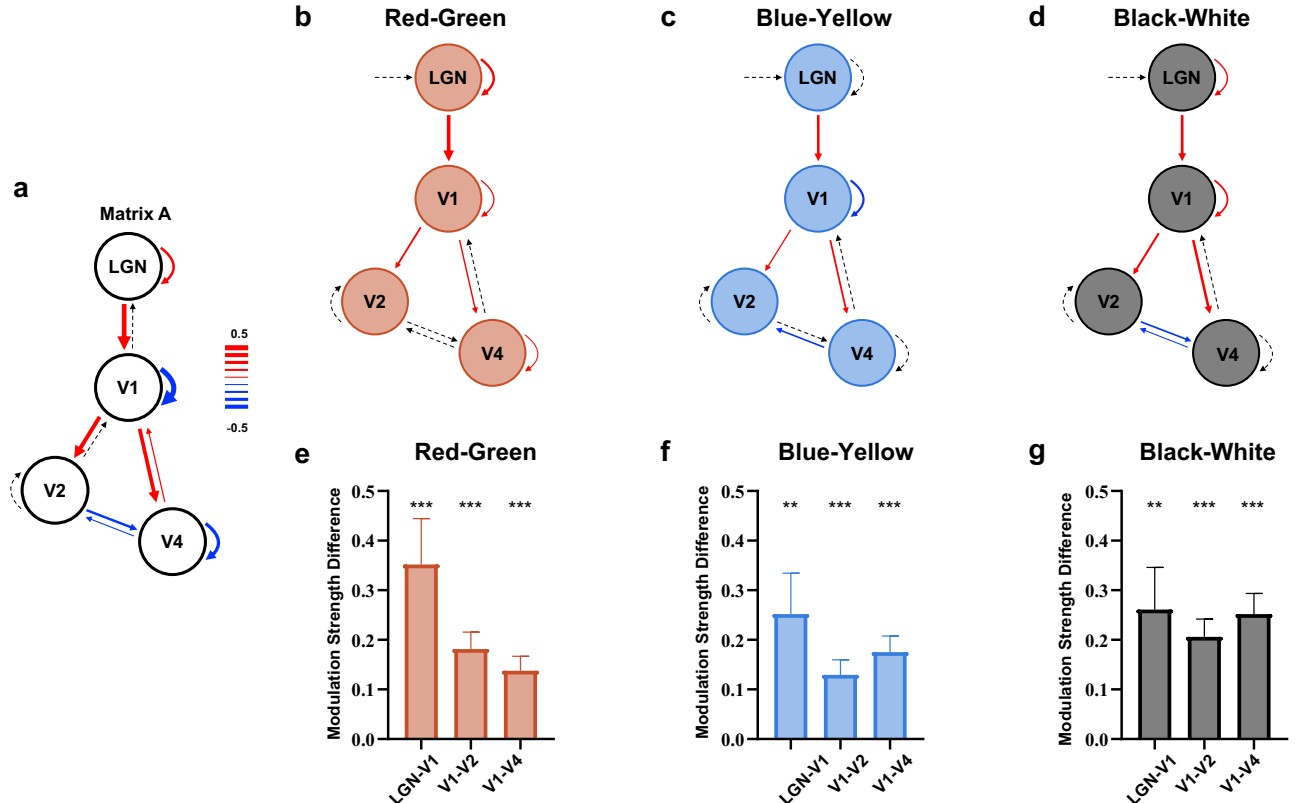

**Fig. 4 | Dynamic causal modeling (DCM) of effective connectivity. a** The full model with all intrinsic connections, in which significant connections (posterior probability > 0.95) are marked in red (excitatory connections) and blue (inhibitory connections), and subthreshold connections are marked with black dashed lines. **b–d** Differences between modulatory connections evoked by red and green, blue, and yellow, and black and white, respectively. The thickness of the lines reflects the size of the difference. Red lines denote more excitatory modulation evoked by the biased colors compared with the unbiased colors, while blue lines represent more inhibitory connections evoked by the biased colors compared with the unbiased colors. Significant differences are shown as solid lines ($p < 0.05$), whereas non-significant differences are shown as dashed lines ($p > 0.05$). **e–g** Differences between the modulatory connections of LGN-V1, V1-V2, and V1-V4 evoked by the two colors in red–green, blue–yellow, and black–white color pairs, respectively. **$p < 0.01$, ***$p < 0.001$, GLMM analysis (two-sided, red–green: LGN-V1, $p = 6.15 \times 10^{-5}$, V1–V2, $p = 1.74 \times 10^{-7}$, V1–V4, $p = 4.36 \times 10^{-6}$; blue–yellow: LGN–V1, $p = 0.003$, V1–V2, $p = 3.26 \times 10^{-5}$, V1–V4, $p = 1.76 \times 10^{-7}$; black–white: LGN–V1, $p = 0.002$, V1–V2, $p = 3.41 \times 10^{-8}$, V1–V4, $p = 1.21 \times 10^{-9}$). Data are presented as mean + SE ($n = 102$ runs). Source data are provided as a Source Data file.

## Laminar distribution of end-spectral bias and polarity bias within V1

The covariance of end-spectral and polarity bias between LGN and V1 together with DCM results in a stronger feedforward drive for dominant color (red and blue), leading to a precortical source of end-spectral bias. In the final session, we further investigated the contribution to asymmetry from three feedforward pathways from the LGN to V1 (magnocellular, parvocellular and koniocellular pathways). We recorded laminar activity in different layers of V1 with a linear multielectrode array (V-probe) in monkeys DS and QQ (Fig. 5a). According to the CSD pattern of the sink and source, V1 can be divided into 6 layers (Fig. 5b). Based on anatomical studies, layers 4 C$\beta$ and 6 receive input from the parvocellular (P) pathway in the LGN; layers 4C$\alpha$ and 6 receive input from the magnocellular (M) pathway in the LGN;[32,33] layer 4 A and the lower part of layer 3 receive input from the koniocellular (K) pathway[12,34–36]; and layers 2/3 and 5, which contain horizontal connections, send output to the downstream visual cortex. If the spectral/polarity bias were pathway specific, then we would find the strongest bias in the input layers for separate pathways.

The laminar distribution of red bias (Fig. 5c) and blue bias (Fig. 5e) showed a decreasing trend from layer 4 to layer 2/3, and the reduced biases were significant between layer 4 (granular, G) and 2/3 (supra-granular, SG) (Fig. 5d, G vs. SG: $p < 0.001$ in a post hoc comparison following one-way ANOVA, $F_{2,263} = 9.9$, $MS_e = 0.16$, $p < 0.001$; Fig. 5f, G vs. SG: $p = 0.003$ in a post hoc comparison following one-way ANOVA,

$F_{2,263} = 5.95$, $MS_e = 0.21$, $p = 0.003$). The strongest bias in the granular layer suggests the dominant role of feedforward input on end-spectral bias, which is consistent with the DCM result of stronger feedforward drive from LGN to V1. The peak locations of red bias and blue bias in layer 4 C$\beta$ indicate a pathway-specific laminar pattern, which coincides with the anatomical destination of the P pathway from the LGN. The decreasing trend from the V1 input layer (4 C$\beta$) to the output layer also corresponded with the decreasing trend of red dominance after V1 observed in the fMRI result (Fig. 3d). In addition, we found a similar decreasing pattern of black bias from layer 4 to layer 2/3, although it was only marginally significant (Fig. 5g, h, G vs. SG: $p = 0.067$ in a post hoc comparison following one-way ANOVA, $F_{2,263} = 3.90$, $MS_e = 0.16$, $p = 0.022$). Furthermore, in contrast to P-pathway-dominant pattern of the red and blue bias, the peak of black bias is located in layer 4 C$\alpha$, which indicates the contribution of the M pathway to black dominance.

To minimize the contamination of the response from luminance-selective cells in the M pathway when evaluating the laminar transition of the color response, we defined another set of stimuli based on the minimal neural response from L4C$\alpha$ and L4B in V1 for each hue in a luminance-matching experiment (Supplementary Note 2) to roughly equate the subjective luminance between the wavelength-varying stimuli (Supplementary Fig. 10). As a result of the control experiment with stimulus luminance matched by response in the M pathway, the laminar distributions of end-spectral bias and polarity bias are very

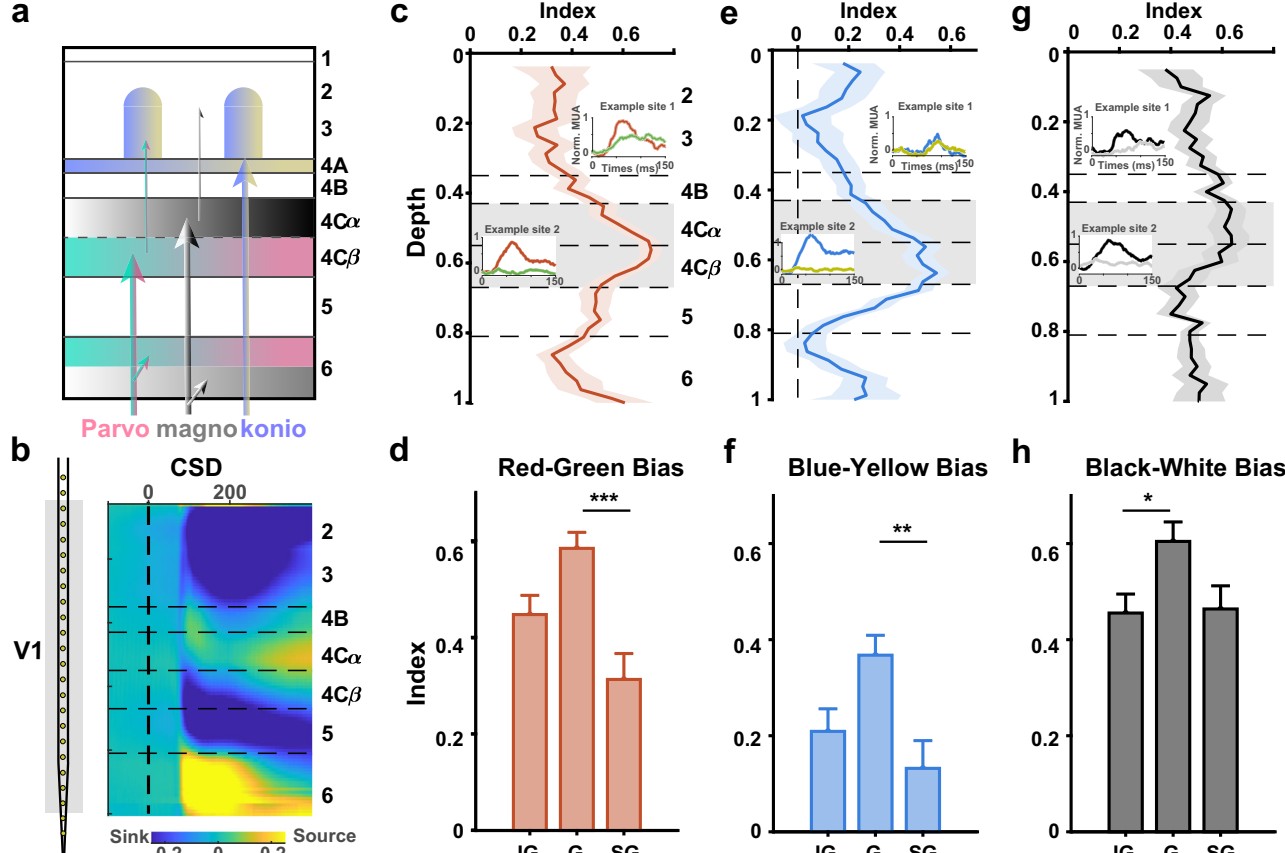

**Fig. 5 | Laminar pattern of response bias in V1. a** Laminar anatomy of the destination of the three pathways from the LGN. **b** Laminar pattern of current source density in V1 after stimulus onset. **c** Laminar pattern of red–green bias in V1. Data are shown as the mean values with standard error. The two insets show the dynamic MUA responses from two example sites at the output and input layers, which were simultaneously recorded with the same probe placement. **d** Red–green bias in infra-granular (including layers 5-6, $n = 94$ channels), granular (layer 4, $n = 89$ channels) and supra-granular layers (including layers 2-3, $n = 83$ channels; IG vs G, $p = 0.064$, G vs SG, $p = 3.82 \times 10^{-5}$). **e–h** Laminar pattern of blue–yellow bias and black–white bias similar to (**c–d**). *$p < 0.05$, ***$p < 0.001$ in Bonferroni-corrected post hoc comparisons following one-way ANOVA (two-sided, n of IG = 94 channels, n of G = 89 channels, n of SG = 83 channels; blue–yellow bias: IG vs G, $p = 0.061$, G vs SG, $p = 0.003$; black–white bias: IG vs G, $p = 0.037$, G vs SG, $p = 0.067$). Data are presented as mean + SE. Source data are provided as a Source Data file.

similar to those in Fig. 5: red bias and blue bias were strongest in L4C$\beta$ (Supplementary Fig. 11, 4C$\beta$ vs. 4C$\alpha$: $p$s < 0.001), while the peak of black bias was located in L4C$\alpha$. The consistent results further confirmed the precortical source and dominant P-pathway contribution of end-spectral bias. For the blue–yellow pair, the strength of the blue bias is bimodally distributed within layers in V1 (Supplementary Fig. 11, one-way ANOVA, $F_{2,245} = 2.80$, $p = 0.06$), with the second peak located in layer 2/3. The bimodal distribution suggests that there might also be a weak contribution of the K pathway to the blue bias.

## Discussion

In this study, we demonstrated the consistency of the end-spectral bias in the same species under the two main measurement techniques – fMRI and electrophysiology. We provided a complete picture of end-spectral bias in the visual cortex (Supplementary Fig. 12), including the neural source and cortical modification, by combining the advantages of large-scale functional imaging and precise laminar recording within V1. First, the correlation between the LGN and V1 suggests a subcortical feedforward source for the end-spectral bias in V1. The color information is further modified by the following cortical stages from V1 to V4, which results in decreasing red bias, increasing blue bias and increasing color representation uniformity along the ventral visual pathway. Next, DCM analysis revealed that the dominant feedforward drive contributed to the end-spectral bias in V1 compared to the feedback modulation from V2 or V4. Finally, by laminar recording

within V1, we found that the red bias in V1 was mainly transmitted through the P pathway and that the blue bias was transmitted through the P and K pathways, which indicates the pathway-specific nature of asymmetric color processing.

The end-spectral bias in V1 was found both in fMRI and in electrophysiological experiments, consistent with previous findings. Stronger spiking activity produced by red/blue than green/yellow was found in the macaque primate visual cortex[6,7,15], especially in the V1 blob, V2 thin strip and V4 columns. End-spectral bias in the visual cortex was also found from 2-deoxyglucose results[8,37] and from intrinsic-signal imaging data[17,25]. Our fMRI results on macaques are also consistent with the human fMRI study[38], which found end-spectral bias in V1 using isoluminant gratings. Here, we used spatially uniform chromatic stimuli in both electrophysiological and fMRI experiments. Based on two-photon calcium imaging data in macaques[26], neurons responsive to such uniform stimuli form clear cluster structures that coincide with CO blobs. From this perspective, the results in this study can be well linked with previous findings in V1 blobs.

The colorimetric parameters of the four hues used in the current study were similar to those used in previous studies[16,17] that reported end-spectral bias in the visual cortex. The mid-spectral colors (green and yellow) were less spectrally pure than the "red" and "blue" colors, and variations in spectral purity (or color contrast) can have a strong influence on physiological response. Although the rank ordering of purity (red = 0.95, blue = 0.94, green = 0.84, and yellow = 0.88) is

similar to the ordering of response strength in V1 of MUA (Fig. 1c) and fMRI (Fig. 1f, except for LFP), the minor difference (0.01) between red and blue seems unlikely to evoke the significant difference in MUA and LFP. Furthermore, the rank ordering of purity is different from the ordering of response in V4. Based on a previous study[8], end-spectral bias still exists after controlling for spectral purity. Therefore, the purity is not the only reason for the end-spectral bias. The consistency of end-spectral bias across the electrophysiological and fMRI evidence on macaque monkeys provided us with the foundation of taking advantage of fMRI to investigate not only the precortical source but also the cortical modification.

The end-spectral bias we found in the LGN is direct evidence to support the precortical feedforward hypothesis for the neural source. The color information relayed by the LGN from the retina flows through anatomically separated connections toward specific layers in V1[36] and leaves some intermixed information toward the thin CO stripe in V2[34]. Although end-spectral bias is well repeated by many studies in V1, whether LGN exhibited the same level of bias, as far as we know, is still an open question. One study that found end-spectral bias in marmoset V1 by intrinsic signals tried to attribute blob activation to cone activity by finding the dominant role of L-M cone contrast that explained blob response in a generalized linear model[16]. A recent MEG study found reduced red dominance on gamma oscillation in the human brain after adjusting color stimuli to create hypothetically matched LGN output[39]. Here, we found highly correlated end-spectral bias between V1 and LGN across sessions, which directly links LGN and V1 based on color asymmetry. In addition to end-spectral bias, various types of color asymmetry have been reported in LGNs, such as normalization strength[40] and contrast sensitivity[41]. The asymmetries of other types in the LGN together with the current finding of end-spectral bias in the LGN commonly suggest a precortical substrate for unbalanced color representation. We also compared the contribution of feedforward and feedback to the end-spectral bias in V1 by DCM analysis, which further strengthened the dominant role of the feedforward mechanism for the generation of coding asymmetry.

Additional direct evidence in this study supporting the feedforward mechanism comes from electrophysiological laminar recording in V1. Red/green information, conveyed by L/M cone activity through parvocellular genuine cells, reaches V1 at layer $4 C\beta$[32,33], where we found the strongest red–green bias of spiking response. Meanwhile, blue/yellow information, conveyed by S/L + M cone activity through koniocellular genuine cells, targets V1 at layer 4 A and the lower part of layer 3, where we also found the second peak of blue–yellow bias (Supplementary Fig. 11, also in Fig. 5e), with the first peak also located in layer $4 C\beta$. Such a laminar pattern of end-spectral bias in awake monkeys is highly consistent with the 2-deoxyglucose uptake pattern of red and blue stimuli from brain sections in an earlier study[8], which indicates the precision of the current laminar alignment. The coincidence between anatomical feedforward connections and laminar patterns suggests the indispensable precortical contribution of the P pathway to end-spectral bias. Furthermore, the laminar pattern of blue bias is surprising. The contribution of parvocellular input layer $4 C\beta$ to blue bias may occur because the blue stimulus evokes substantially stronger responses along the L-M cone contrast axis (0.97) than the yellow stimulus does (−0.04), which is similar to red (0.74) overtaking green (−0.23) along the L-M cone contrast axis. Such cone-contrast consistency between the blue–yellow pair and red–green pair was consistent with a previous study[16]. End-spectral bias (larger response to red and blue stimuli than green and yellow ones) reflects the larger cone contrasts for the stimuli used in the measurements. It is quite possible (perhaps likely) that the end-spectral bias arises from simple subtractive combinations of the three different photopigments in the retina, although this has not yet been extensively tested.

By taking advantage of the combination of laminar recording and fMRI, we further demonstrated that the red bias in V1 is even stronger than that in LGN, especially in the target layer of the P pathway. The increasing red bias from the LGN to V1, indicated by both the fMRI results and the laminar patterns, suggests that the red bias is magnified during the projection from the LGN to V1 instead of being magnified through the feedback connection in cortical areas. This result indicates that the LGN is not the only contributor to the end-spectral bias in V1 and that LGN-V1 projections to $L4C\beta$ also enhance such bias (the contribution of Parvo cells in V1 is nonlinear).

Within V1, we were surprised to find a significant laminar transition of color asymmetry, especially for red–green opponency. The red bias was significantly reduced from the input layer ($4 C\beta$) to the output layer (2/3), which suggests that the color processing in V1 was not a purely passive feedforward mode for bias from the subcortical upstream area. There may be two mechanisms for the laminar change in red bias. On the one hand, as previous studies suggest, V1 is the first place where mixing between the cone-opponent mechanism occurs[42]. The transition of red bias within the V1 column may reflect the mixing process, which leads to the shift of hue preference to new mingled ones. Through secondary transsynaptic connections from layer $4C\beta$, L/M opponent signals arrive at the superficial layer in V1[34], where they meet S cone information through the K pathway. At this stage, the opponent signals grouped segregated neurons with different color preferences[17,25,26]. On the other hand, the reduced red bias in the superficial layer may be due to stronger cortical inhibition for red color. The superficial layer in V1 includes long-range horizontal connections, which help communicate homogeneous and heterogeneous color information across cortical space[43]. The smaller neuron ensemble for green located in the middle of the blob appears to be nonoriented[25], which may lead to weaker surround suppression[24] than the larger neuron ensemble for red, which is mostly located in the interblob. The mixing mechanism within columns and horizontal inhibition across the cortical surface may jointly contribute to the reduction in red bias.

The reduced red bias within V1 further decreases along the ventral visual pathway from V1 to V4. The consistent decreasing transition within V1 and beyond V1 suggests a common cortical mechanism in V1 and in the higher visual cortex, as described above. Such a transition was consistent with previous reports from imaging data in macaque monkeys and humans. Contrary to the end-spectral bias in human V1, human V2/V3 (thick stripes) showed a stronger response to mid-spectral hues, while responses in higher areas to all four hues were statistically equivalent[38]. In monkeys, the pattern of end-spectral dominance receded in V2 thin stripes and was absent in V4[17]. In the present study, due to the limitations of the spatial resolution in our study (1.5 mm isotropic), it was challenging to reveal and investigate these stripes [reported to be 1–1.5 mm in macaques[44]] or pathway-specific layers within V1 (L4Cα and L4Cβ). Our results may be predominantly influenced by thin stripes due to the use of uniform square stimuli in our experiment, in contrast to the drifting gratings employed in previous investigations (thick stripes and interstripes are sensitive to moving stimuli). Future studies with higher spatial resolution and specific methodologies may be needed to further investigate and compare the color bias across stripes. The highly segregated horizontal patches in V1, V2 and V4 together with the V1 laminar structure for color processing also suggest a possible laminar difference between different stripes in V2, which compounds the future need for multiple-technology combinations. Neurophysiological laminar recordings in V1, V2 and V4 guided by imaging results of stripe locations are indispensable to uncover the possible dissociated pathway of color representation between patch and interpatch regions in V2 and V4.

Unexpectedly, we found an increasing trend of blue bias from LGN to V4. This occurred possibly because the S-cone influence may be stronger in higher visual areas[14]. Based on the laminar distribution of blue bias, a stronger blue response results from the P pathway (L/M-cone contribution) and K pathway (S-cone contributions). A previous

single-cell recording study found that the contribution of the S-opponent path is doubled at V1 relative to that at the LGN[45]. Therefore, the blue bias may be enhanced through stronger feedforward projection from the LGN to V1, which is similar to the red bias. Our DCM result was also consistent with this explanation. At the cortical stages, due to the layer specification of the K pathway in V1, only deep layers 3 and 4 A receive S/(L + M) signals that can project directly to V2[36], which may lead to roughly weak activation for V1 imaging in fMRI.

Note that the current findings were obtained for neurons/voxels with receptive fields ranging from 2–6 degrees of visual angle in eccentricity. Within such receptive fields, we did not observe significant patterns of change depending on eccentricity (see Supplementary Fig. 13). However, it remains unclear whether the end-spectral biases also exist in other eccentric regions and whether there are differences in the end-spectral bias among different eccentric regions. Although no studies have directly investigated the end-spectral bias across regions with different eccentricities in V1, V2, and V4, previous evidence has reported no variation in color sensitivity (at least between 0° and 10°) in V1[8]. Notably, a previous study[17] conducted in macaques demonstrated that when full-screen stimuli were used and signals were recorded in regions corresponding to the peripheral eccentricity at V1, V2, and V4, homogeneity among different hues gradually increased from V1 to V4, consistent with our findings within the receptive field range of 2-6 degrees. Taken together, the evidence suggests that the trend of change in the end-spectral bias from LGN to V4 and the end-spectral bias within these regions might remain relatively stable within the receptive field range of 2-6 degrees in eccentricity and even outside this range, which might be interesting to investigate in future studies.

Overall, greater homogeneity in higher visual cortical areas may reflect shifts away from cone-opponent coding in the LGN and V1. The different transitions through the visual hierarchy between red–green and blue–yellow opponency indicate that the color processing after V1 can still be pathway specific. A recent study found that high intraocular pressure may lead to stronger impairment of blue–yellow responses than red–green responses in V1, V2 and V4[46]. Temporal S-cone information declined more steeply than temporal L-M information from V1 to higher areas (V2 and V4)[47]. Here, we further illustrate that the coding asymmetry in the K pathway grows slower and reaches a peak later through visual hierarchy than the asymmetry in the P pathway: red bias reaches a peak in V1, while blue bias continues to increase after V1. On the one hand, the slow increasing blue bias in the higher visual cortex may reflect the evolution of nocturnal habits to diurnal habits because short-wavelength blue color is the strongest synchronizing agent for the circadian rhythm[48]. On the other hand, the asymmetry in blue–yellow color processing may contribute to primate perception bias, for example, a famous "color constancy" phenomenon in the color of "the dress"[49]. Future studies may further investigate the neural correlate of blue bias and its functional influence on color perception.

Taken together, our findings demonstrated a feedforward drive for end-spectral bias from the LGN, which is also illustrated by the network modeling for the LGN, V1, V2 and V4. Such end-spectral bias was conveyed by the feedforward parvocellular and koniocellular pathways from the LGN to V1. The highly consistent results in three dimensions − fMRI, electrophysiology and modeling − provide a comprehensive perspective of the precortical generation of coding asymmetry for surface color and help to uncover the cortical modification of the color asymmetry that tends toward uniformity through visual hierarchy.

## Methods
### Electrophysiological experiment
#### Preparation of awake monkeys
All procedures were conducted in compliance with the National Institutes of Health Guide for the care and use of laboratory animals and were approved by the Institutional Animal Care and Use Committee of Beijing Normal University. Four male adult rhesus monkeys (DS, DQ, QQ, Macaca mulatta, 5−7 years old, 4−8 kg, DP, Macaca fascicularis, 6 years old, 8 kg) were used. Under general anesthesia induced with ketamine (10 mg/kg) and maintained with isoflurane (1.5−2.0%), a titanium post was attached to the skull with bone screws to immobilize the animal's head during behavioral training. For monkeys DS and QQ, after the animal had been trained with a simple fixation task, a circular titanium chamber (20 mm in diameter) with a removable lid was fixed over the craniotomy (15 mm anterior to the occipital ridge and 14 mm lateral from the midline) with dental cement for chronic recordings from V1 and V2. For monkey DQ and DP, arrays were implanted in the V1 area (same region as the chamber location of monkey DS and QQ). After the signal decayed with time from the first array of DQ in the right cortex, a second array was implanted next to the first array. Antibiotics and analgesics were used after the surgery.

#### Visual stimulation
Visual stimuli were generated with a stimulus generator (ViSaGe; Cambridge Research Systems) under the control of a PC running a custom C11 program developed in our laboratory. The stimuli were displayed on a 22-inch CRT monitor (Dell, P1230, 1200 × 900 pixels, mean luminance 45.8 cd/m², 100 Hz refresh rate). The CRT monitor was gamma corrected (gamma = 2.05) and calibrated by LightScan software equipped with OptiCAL Photometry (Cambridge Research Systems). The viewing distance was 114 cm.

Square presentation was used to measure color and luminance response (4° × 4°). In the formal experiment, 6 types of squares were used (red, green, blue, yellow, black, white). The center or edge of a square stimulus located in the RF in each trial. For the Utah array experiment, there were five stimulus positions: the left, right, top, and bottom edges of a square in the RFs and the center of the surface in the RFs (Supplementary Fig. 1). For the linear probe experiment, the center surface of a stimulus was located in the RFs. The four hues (dominant wavelength: red = 605 nm, green = 549 nm, blue = 466 nm, yellow = 567 nm) were equiluminant to the gray background (~10.6 cd/m² in electrophysiology experiments). To generate isoluminant hue stimuli, we used a spectroradiometer (PR-670 SpectraScan, Photo Research Inc.) to measure the luminance of a series of gradient stimuli (red, green, blue, yellow, gray) presented by the CRT monitor. We then fitted the curves to find the RGB values of each stimulus (red, green, blue, yellow, gray) whose luminance was equal to the maximum luminance that blue can achieve. In this way, we ensured that all four colors and the gray background were isoluminant to the chosen RGB ratio. The black stimuli were generated by setting the RGB values to 0, and the black and white stimuli had equal contrast to the background. We measured the real parameters of the CIE1931 color space with the spectroradiometer and further calculated the cone excitations and cone contrasts of all stimuli from CIE coordinates and luminance[50]. The cone contrast for a particular cone class here is defined as the ratio the cone excitation due to the square stimulus relative to the cone excitation due to the background. We provide the parameters of the stimulus in multiple color spaces in Supplementary Table 1 and show the cone excitations in the MacLeod-Boynton chromaticity diagram in Supplementary Fig. 14.

Sparse noise was used to simultaneously map the receptive fields (RFs, see details below). Random orientation presentation was used to measure orientation dynamics, align laminar positions, and check the verticality of the probe[51].

#### Behavioral task
In the fixation task, a trial began when a monkey began fixating on a 0.1° fixation point (FP) presented on a CRT screen. In each trial, the FP was displayed in the center of the screen. The animal's eye positions were sampled at 120 Hz using an infrared tracking system (ISCAN). Within 300 ms of FP presentation, the animal was required to fixate within an invisible circular window (1° in radius)

around the FP. Then, a trial was initialized. After the monkey maintained fixation on a blank screen for 400 ms, the square stimulus was displayed for 2 s, followed by a blank interval of 400 ms. The FP then disappeared, and the animal received a drop of water as a reward. A trial was aborted if the animal's fixation moved outside the fixation window. In the sparse noise experiment and random orientation experiment, the stimulus sequence was displayed for 3 s.

**Electrophysiological recording.** We simultaneously recorded neuronal activity in V1 using a linear probe (V-probe, Plexon; 24 recording channels spaced 100 μm apart, each 15 μm in diameter) and three arrays (Blackrock, 10 × 10, length 1 mm, interchannel spacing 400 μm). The linear array was controlled by a microelectrode drive (NAN Instruments), and the depth of each probe placement was adjusted to extend through all V1 layers. The raw data were acquired with a 128-channel system (Blackrock Microsystems). The raw data were high-pass filtered (seventh-order Butterworth with 1000 Hz corner frequency), and multiunit spiking activities (MUAs) were detected by applying a voltage threshold with a signal-to-noise ratio of 5.5. The raw data were also low-pass filtered (seventh-order Butterworth with 300 Hz corner frequency) to obtain LFPs. Both MUAs and LFPs were downsampled to 500 Hz.

**RF mapping.** After manually mapping the RFs of the recording channels, we used sparse noise to identify the precise RF center. The sparse noise consisted of a sequence of randomly positioned (usually on a 13 × 13 or 11 × 11 sample grid) dark and bright squares (0.1° – 0.3°, contrast 0.9) against a gray background (luminance 45.8 cd/m²). Each sparse noise image appeared for 20 ms with at least 50 repetitions. The sequence was cut into small segments based on the trial length. We obtained a two-dimensional map of each channel. Responses averaged from the x- and y-axes of each map were fitted with a two-dimensional Gaussian function to estimate the center position and radius of each RF ($2\sigma$ of Gaussian function). The RFs were located within 5° of the fovea.

**Normalization.** All data analyses were performed using custom programming in MATLAB (MathWorks). We normalized the data by the peak response within 40–200 ms after stimulus onset after averaging the responses from all conditions.

**Spectral analysis.** To obtain the raw power spectral profile of the LFP response in Fig. 1, we used a multitapered Fourier transformation[52] [time–bandwidth product to estimate tapers, 3; tapers, 5; Chronux toolbox (http://chronux.org/)]. To quantify the response of the gamma component in terms of different colors, we used a simplified one-dimensional descriptive model to fit the raw power spectrum, which is similar to the two-dimensional model in our previous study[53].

$$Power(f) = Baseline(f) + \sum_{i=1}^{3} Comp_i(f) \quad (1)$$

$$Baseline(f) = \frac{k}{f^a + b} + c \quad (2)$$

$$Comp_i(f) = w_i \cdot e^{-\frac{(f-\mu_i)^2}{2\sigma_i^2}} \quad (3)$$

**Laminar alignment.** To align different probe placements in depth, we used the laminar pattern of MUA responses combined with the current source density (CSD) analysis[54,55] of LFP signals[22,51]. The CSD profile can be estimated according to the finite difference approximation, taking the inverse of the second spatial derivative of the stimulus-evoked

voltage potential φ, defined by:

$$CSD(x) = \frac{\varphi(x+h) + \varphi(x-h) - 2\varphi(x)}{h^2} \quad (4)$$

where x is the depth at which the CSD is calculated and h is the electrode spacing (100 μm).

## fMRI experiment

**Subjects and general procedures.** Four male macaque monkeys (monkeys Q, P, N, and B; *Macaca mulatta*; 9-10 y old; 7.5–11.5 kg) were used. They were acquired from the same primate breeding facility in China, where they had social group histories as well as group-housing experience until their transfer to the Institute of Biophysics, CAS (IBP) at the age of approximately 4 y. After that, they were individually caged with auditory and visual contact with other conspecifics in the same colony room, which accommodates approximately 10 rhesus monkeys. All animals used in this study had been housed at IBP for 5–6 y before this experiment. All experimental procedures complied with the US National Institutes of Health Guide for the Care and Use of Laboratory Animals and were also approved by the Institutional Animal Care and Use Committee of IBP. Each monkey was surgically implanted with a magnetic resonance (MR)-compatible head post under sterile conditions using isoflurane anesthesia. After recovery, subjects were trained to sit in a plastic restraint chair and fixate on a central target for long durations with their heads fixed, facing a screen on which visual stimuli were presented[56,57].

**Visual stimulation.** The stimuli were presented using Presentation software (version 21.1, www.neurobs.com) and projected on a translucent screen with an MRI-compatible projector at a resolution of 1024 × 768 @ 60 Hz. The viewing distance for monkeys was 68 cm. Six stimulus sets were used: red, green, blue, yellow, black, and white squares. For each stimulus set, there were two types of configurations: the square aligned to the left/right horizontal meridian (see examples in Fig. 1e). Each stimulus includes two squares sized 4 × 4° of the same color with its inner edge 2 degrees from the central fixation square.

The four colored stimuli (red, green, blue, yellow) were equiluminant to the gray background ( ~ 5.2 cd/m² in fMRI experiment). The process for defining the isoluminant hue stimuli was the same as in electrophysiological experiments: we measured and fitted the luminance of a series of gradient stimuli (red, green, blue, yellow, gray) presented by the projector and found the RGB values of each stimulus corresponding to the maximum luminance that blue can achieve. The black stimuli were generated by setting the RGB values to 0, and the black and white stimuli had equal contrast to the background. We also measured the real parameters of the CIE1931 color space by the spectroradiometer and further calculated the cone excitations and cone contrasts of all stimuli from CIE coordinates and luminance[50] (Supplementary Table 1), and we present the cone excitations in the MacLeod-Boynton chromaticity diagram in Supplementary Fig. 1.

**Experimental design and task.** We presented 6 stimulus sets (12 stimuli) to each monkey in a rapid event-related fMRI experiment. Stimuli were presented for 2 s on a constantly visible uniform gray background. A fixation square (0.4°) was always presented at the center of the screen. Each stimulus was presented 5 times in each run, during which there were a total of 26 randomly interspersed null trials with no stimulus presented. Individual scanning runs began and ended with three null trials. The trial-onset asynchrony was 5.56 s (4 TRs). The stimulus onset asynchrony was either 5.56 s or a multiple of that duration when null trials occurred in the sequence. Each run lasted 7 min and 58.16 s. The trials (including stimulus presentations and interspersed null trials) occurred in random order (random without replacement). Different random sequences were used in each run.

Eye position was monitored with an infrared pupil tracking system (ISCAN, Inc). In the current experiment, the monkeys were required to maintain fixation on a square in the center of the screen to receive a liquid reward. In the reward schedule, the frequency of reward increases as the duration of fixation increases[56,57]. The four monkeys were scanned in three to five separate sessions each. Data were included from only those runs/trials in which fixation was maintained well [mean (SE): 88.08% (0.86%)], resulting in a total of 20-30 runs (85-143 stimulus repetitions) per monkey.

**Brain activity measurements.** Functional and anatomical MRI scanning was performed in the Beijing MRI Center for Brain Research (BMCBR). Before each scanning session, an exogenous contrast agent [monocrystalline iron oxide nanocolloid (MION)] was injected into the femoral or external saphenous vein (8 mg/kg) to increase the contrast/ noise ratio and to optimize the localization of fMRI signals (Leite et al.[58]). Imaging data were collected in a 3 T Siemens Prisma MRI scanner with a surface coil array (eight elements). Twenty-seven 1.5 mm coronal slices (no gap) were acquired using single-shot interleaved gradient-recalled echo planar imaging. Imaging parameters were as follows: voxel size: 1.5 mm isotropic, field of view: 129 × 129 mm; matrix size: 86 × 86; echo time (TE): 17 ms; repetition time (TR): 1.39 s; flip angle: 90°. A low-resolution T2 anatomical scan was also acquired in each session to serve as an anatomical reference (0.625 mm × 0.625 mm × 1.5 mm; TE: 101 ms; TR: 11.200 s; flip angle: 126°). To facilitate alignment to the template, we also acquired high-resolution T1-weighted whole-brain anatomical scans in separate sessions. Imaging parameters were as follows: voxel size: 0.5 mm isotropic; TE: 2.84 ms; TR: 2.2 s; flip angle: 8°.

**fMRI data analysis.** Functional data were preprocessed using Analysis of Functional NeuroImages software (AFNI)[59]. Images were realigned to the volume with the minimum outlier. Then, the data were smoothed with a 2 mm full-width half-maximum Gaussian kernel. Signal intensity was normalized to the mean signal value within each run. For each voxel, we performed a single univariate linear model fit to estimate the response amplitude for each condition. The model included a hemodynamic response predictor for each category and regressors of no interest (baseline, movement parameters from realignment corrections, and signal drifts). A general linear model and a MION kernel were used to model the hemodynamic response function. All fMRI signals throughout the study were inverted so that an increase in signal intensity indicated an increase in activation. To perform the group analyses, we aligned statistical map images onto images of the symmetric NIMH Macaque Template (NMT) v2[60].

**Definition of regions-of-interest (ROIs).** For each monkey, the set of visual responsive voxels was defined as the voxels that were more active during the stimulus presentation than during the baseline (p < 0.001) and were restricted to the anatomically defined areas of LGN, V1, V2, and V4[60–62]. For each stimulus configuration, in which there were two squares, we defined two sets of sub-ROIs (one on the left hemisphere and another on the right hemisphere, corresponding to two small squares). That is, in total, there were four sub-ROIs in one brain area (e.g., V1), corresponding to the four locations of squares. For each sub-ROI, based on the averaged responses across all six stimuli, the top 20 visual responsive voxels were selected, and then voxels far away from the center of surviving voxels were removed to yield the final sub-ROIs [mean(SE): 18.46(0.32) voxels] (V1: Fig. 1e; LGN: Fig. 2a; V2&V4: Fig. 3a). The signal was extracted from these sub-ROIs. For each stimulus set, the average responses across the four sub-ROIs were calculated within each run.

Based on discussions on circularity in a previous study[63], the current ROI definition and analyses should be impacted by circularity to a limited degree. To further address the issue of circularity and to provide reassurance regarding the robustness of our results, we conducted an additional analysis using a stricter approach. We split the data into two sets: odd and even runs. We defined the ROIs using only one-half of the data (e.g., odd) and examined the responses on the other half (e.g., even). Then, we repeated this process in reverse, using the even runs to define ROIs and examining the responses in the odd runs. Finally, we averaged the results obtained from both steps. The results from this analysis demonstrated a similar trend to our current findings, albeit with slightly reduced significance due to the reduced sample size (Supplementary Fig. 15).

For each ROI set, we calculated the index of bias for each pair of stimuli (i.e., red versus green, blue versus yellow, and black versus white) following Eq. 5. To qualitatively investigate the representation uniformity on neural response across all four basic colors, the homogeneity index was also calculated as the reciprocal of the standard deviation multiplied by the mean value of color responses for each ROI (Eq. 6). When we combined the bias indices from different subjects/ sessions, to account for individual differences and intersession differences due to factors such as MRI coil placement and contrast agent clearance, the fMRI signals evoked by different stimuli within each run were normalized to 0-1 for each session (Eq. 7).

$$Index = \frac{colorA - colorB}{|colorA| + |colorB|} \tag{5}$$

$$Homogeneity\ index = \frac{mean(red, green, blue, yellow)}{std(red, green, blue, yellow)} \tag{6}$$

$$Normalized\ x_{i,j} = \frac{x_{i,j} - min(X)}{max(X) - min(X)} \tag{7}$$

where $x_{i,j}$ is the averaged responses evoked by one color (i) within one run (j).

Note that the contrast-to-noise ratio [CNR, calculated as responses (A) divided by the noise variance ($\sigma_N$), Eq. 8] may impact the homogeneity index. Therefore, we evaluated the CNR to measure the activation fluctuations relative to the noise in our fMRI experiment[64]. To further address whether the CNR might impact the homogeneity index, we treated the CNR as a covariate and reanalyzed the data (Supplementary Fig. 4b).

$$CNR = \frac{A_{(allstimuli)}}{\sigma_N} \tag{8}$$

**Dynamic causal modeling.** We used dynamic causal modeling (DCM)[65] implemented in SPM12 to evaluate the modulation exerted by different stimuli. More specifically, the DCM parametric empirical Bayes (PEB) approach was performed[66,67]. Briefly, PEB is a hierarchical Bayesian model that uses both nonlinear (at the first level) and linear (at the second level) analyses. First, relevant time series were extracted, preprocessed, and summarized within each sub-ROI in each brain area (i.e., LGN, V1, V2, and V4) by performing a principal components analysis (PCA) and retaining the first principal component as in previous guidelines[68]. This preprocessing step involved extracting the principal eigenvariate, which effectively captured the data characteristics beyond the average amplitude changes typically reflected by the mean value. By doing so, the impact of response variations across experimental conditions was mitigated. For the first level, the full DCM model was estimated separately for each run using the variational Laplace estimation scheme[69], which allows us to offer the best trade-off between model accuracy and complexity. Then, the second (group) level analysis used the PEB module implanted in SPM (spm_dcm_peb and spm_dcm_peb_bmc).

We defined a full model (Matrix A) based on known anatomical connections of the primate visual system[70,71]. The driving input was restricted to the LGN. We set bidirectional endogenous connections between the LGN and V1, V1 and V2, V1 and V4, and V2 and V4. Moreover, all self-connections were switched on, which are unitless log scaling parameters scaled up or down the default value of −0.5 Hz. Thus, the more positive the self-connection parameter, the less it will respond to the inputs from the network. The values of other connections in matrix A represent the rate of changes in the destination region affected by the source region, in units of Hz. Matrix A represents the effective connectivity between nodes, while Matrix B shows the changes in the effective connectivity due to the modulatory inputs. The values in matrix B represent the changes in the coupling from the source region to another destination region caused by the experimental input. Of note, positive values indicate excitatory influences, whereas negative values indicate inhibitory influences. The values of self-connections in the B matrix were inverted so that more positive values indicate that the ROI responds to the stimuli more sensitively.

**Statistical analysis.** Because of the small sample size, a typical fixed effects model for monkey fMRI studies was used. To compare the signal changes for each color pair, we performed generalized linear mixed model (GLMM) analyses. Color (colors A and B in one color pair) was treated as a fixed factor, and monkeys, sessions, and runs were set as random factors. GLMMs were also performed on the end-spectral and polarity bias to compare the biases from the LGN to V4 with the ROI (LGN, V1, V2, V4) treated as a fixed factor, and random factors were the same as above. Likewise, we also performed GLMMs on the DCM results. We then followed up with post hoc tests with adjustments for multiple testing using the Bonferroni method. We note here that all p values are corrected unless specified otherwise. Two-tailed Spearman correlation analysis was performed to calculate the correlation between the indices of end-spectral and polarity bias in the two studied ROIs. The variable of individual monkey identity was controlled by converting it into binary values (dummy variables)[72].

### Reporting summary
Further information on research design is available in the Nature Portfolio Reporting Summary linked to this article.

## Data availability
The raw data that support the findings of this study are available from the corresponding author upon request. Source data are provided with this paper.

## Code availability
The source code to reproduce results of this study is available from the corresponding author upon request.

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

## Acknowledgements

This study was supported by STI2030-Major Projects (2022ZD0204600 [D.X.]), the National Natural Science Foundation of China (grant no. 32171033 [D.X.]), STI2030-Major Projects (2021ZD0200200 [N.L.]) and STI2030-Major Projects (2021ZD0204200 [M.Z.]). We thank Tianshu Yang and Zhaojin Cheng for the animal surgery, and Zhentao Zuo for fMRI technical assistance.

## Author contributions

Y.W. and M.Z. contributed equally. Y.W., M.Z., N.L., and D.X. designed the research. T.W., J.H., Y.W., X.S., N.L., and D.X. performed the surgery. Y.W. performed the electrophysiological experiments and collected data. Y.W. analyzed electrophysiological data. M.Z., H.D., Y.X., and N.L. performed the fMRI experiments and collected data. M.Z. analyzed fMRI data. M.Z. and H.D. performed animal behavior training. Y.W. and M.Z. wrote the draft. Y.W., M.Z., D.X., N.L., H.D., T.W., Y.X., T.Z., W.D., J.H., and X.S. reviewed and edited the paper.

## Competing interests

The authors declare no competing interests.
