## [Peer Review File · Nature Communications]

The neural origin for asymmetric coding of surface color in the primate visual cortexREVIEWER COMMENTS

Reviewer #1 (Remarks to the Author):

General

The aim of the manuscript is to understand whether apparent biases for particular colors in visual cortex emerge in visual cortex, or are already present in the input to cortex. To address this the authors combine MION-enhanced fMRI measurements from LGN, V1, V2 and V4 of 4 awake macaque monkeys, with measurements of local field potential and multiunit activity (made using multichannel probes or arrays) from V1 of another 3 awake macaque monkeys.

In each case the major analyses focus on responses to a colored stimulus. This was a 4x4 degree square presented just outside the fovea, with stimuli one of red, green, blue, yellow, black, or white flashed for a couple of seconds and responses recorded. The authors find stronger LGN and V1 fMRI responses for red (vs green), blue (vs yellow) and black (vs white), and similar biases for multiunit activity and gamma LFP power in V1. The authors conclude that the color bias in V1 is largely inherited from LGN, but may be reinforced by processing in V1.

Overall, the manuscript provides a rare comparison of fMRI and electrophysiological measurements under similar viewing conditions in awake monkeys. The analyses of fMRI and electrophysiological data appear to be generally appropriate. The results are consistent with the idea that some of the biases seen in visual cortex reflect biases already generated in the retina or LGN, which is also consistent with previous work on color responses in monkey.

My major concern is that the color of the stimulus is very poorly described and it is currently impossible to reproduce the experiments, because the intensity and color of the stimuli are not provided. The methods state that the stimuli were equiluminous, but it is not clear whether this means the colored stimuli (red, green, blue, yellow) were equiluminous to the background, or whether they were equiluminous to each other but different intensity to the background. I assume it is the former, but this needs to be spelled out. Indeed very few details are provided: only that the electrophysiology experiments were conducted on a CRT monitor, at 45.8cd/m²; and that the fMRI experiments were conducted on a MRI-safe projector. No details of the calibration are given (either gamma calibration of the CRT or projector primaries, or methods for color-calibration). At a minimum the manuscript needs to include colorimetric definitions of the stimuli (e.g. CIE specifications). The authors also need to calculate the cone-contrast of each of the stimuli, and report those cone contrasts. Without knowledge of the cone contrasts, some of the data are not possible to interpret. The contribution of parvocellular input layer 4Cbeta to blue bias (Fig 5E) is a case in point – is this because the blue stimulus evokes substantial

responses from red or green cones, or because blue cones provide an input to the parvocellular pathway?

Other points

Line 51: At this stage it is not clear especially for a non-expert why the representation of red- and blue relative to green/yellow can be likened to the 'black dominance'. Would recommend removing.

Lines 53-58: this is a long sentence and seems to imply that neurons in blobs are responsible for stronger gamma oscillations – I don't think there is evidence for that per se. Would recommend removing moving comments about blobs to discussion and restricting introduction to generic cortical circuits as in the subsequent paragraph.

Line 60: There are a few things going on in this paragraph. I think generally that what you are saying is simply that bias for particular colors may arise because the input to cortex has that bias, or because of the way that cortex its inputs. However, this is muddled somewhat because many of the comments about inhibition and recurrence either need to be spelled out more or omitted. It's not clear to me that they add to the main point here – could be better off simply saying that you want to establish this, by characterising the input, and the cortical processing, as you do in the final paragraph.

Line 67: not sure what you mean by 'high density' of neurons preferring red- and blue. Do you simply mean that there are more of them in the superficial layers than neurons preferring other colors?

Line 89: you say "one monkey" for linear probes but note both DS and QQ used here and in methods. You then say "the three monkeys were trained" on line 92 but note 4 animals in Figure 1 legend. This is all a bit confusing – please clarify.

Line 93: In fMRI methods on line 506 you state a 2s stimulus epoch, here it is 3s. Please clarify (and make in clear in both electrophysiology and fMRI methods sections).

Line 94: You say that the in some cases you made the stimulus 'cover half RF by stimulus edges'. I am not sure I understand this – an image showing the RFs of the neurons, and the outline of the stimulus, would be useful.

Line 95: I am not sure what you mean by 'ensured that the neural response to the equal-luminance stimulus was lowest'). Please clarify.

Line 156: I am not sure what the correlations in Fig 2C-E imply. I think that each point represents one run from one animal, but its not clear to me why individual runs are been shown here. The unit of observation would seem to be the animal, so analysis that averaged runs within animals, or somehow took account of animal as a grouping factor would seem to be important to be able to show that bias within LGN predicts bias within V1, or vice versa.

Line 177 and around it: The analyses suggest a difference between V1 and V4 (and less, if any difference between V1 and V2, or V2 and V4) in color bias. Comparisons of responses in V1, V2 and V4 are made difficult by the fact that the retinotopy in the three regions is quite different. The ROIs appear to be based on the voxels that were responsive to each of the two visual stimulus locations, introducing possible circularity. I would like to see some analysis of voxel response that took into account the possibility that color bias depends on eccentricity, and asked whether changes in V4 survive such analysis.

Lines 190-193. I don't actually understand this part. Could you find other ways to state this to make it easier?

Line 231: Is it possible that the greater feedforward strengths for red, blue, black simply reflect stronger responses to those colors? It would be useful to distinguish.

Figure 5: It would be useful to see some examples of the responses that go into making the line graphs in panels C, E, & G.

Line 323: many other groups have reported asymmetries of various sorts in LGN and in V1.

Line 402: This seems to imply recordings from 3 animals, but include 4 – reword to make clearer the total number of animals.

Line 412: 'After the signal dying out' – I understand what you mean, but this should be made clearer.

Reviewer #2 (Remarks to the Author):

Review of “The neural origin for asymmetric coding of surface color in the primate visual cortex” by Wu, et al.

General Comments

This manuscript employs multiple data acquisition techniques to thoroughly demonstrate the prevalence of the end-spectral bias throughout the ventral visual pathway in non-human primates, ultimately building a coherent case for a subcortical origin and a feedforward progression of this well-studied neural coding privilege. The strongest evidence being the laminar response pattern in V1, with peaks predominantly found in the granular input layer, and the consistent color pairing response differences (end-spectral bias) across both invasive and non-invasive recording techniques. The substantially weaker (R-G, B-W) or nonsignificant (B-Y) end-spectral bias findings in LGN compared to the cortical ROIs, the potential for fSNR differences across ROIs, and the stronger LGN bias correlations with extrastriate visual areas compared to the striate visual area (V1) adds some uncertainty to the feedforward conclusions being drawn.

More broadly, this study strongly reinforces the link between neurophysiology and functional neuroimaging (fMRI) techniques by demonstrating qualitatively similar results across these experimental domains while using an identical behavioral paradigm. Additionally, it provides a straightforward framework for reproducing and building upon the reported findings further with human neuroimaging experiments.

There are a few minor points related to the methods which require further clarification, but in general the methods provide adequate detail for the work to be understood and reproduced. There are several concerns related to how certain results are reported and interpreted which requires some further explanation and revision in order to fully support the main conclusions of the study.

Methods

1. There are inconsistencies in the reported number of monkeys used in the electrophysiology experiments. At multiple points in the results and methods sections it is mentioned that three monkeys were used for data collection, but subsequently the IDs for 4 monkeys are reported.

2. When describing the RF mapping procedure, the (Michelson?) contrast level of the sparse noise stimulus is reported in units of degrees, instead of percentage. Furthermore, it is unclear if the “0.1° - 0.3°” actually refers to the range of visual angle which the sparse noise stimulus occupied, or is somehow related to the contrast level.

3. Within the fMRI data analysis section, it states that “All fMRI signals throughout the study were inverted so that an increase in signal intensity indicates an increase in activation”. Further explanation and justification for this uncommon fMRI preprocessing step should be provided as it is unclear why it was necessary. Is this inversion necessary when using the MION contrast agent?

Results

1. When reporting the results presented in Figure 1G (last paragraph before the “End-spectral and polarity bias in LGN” section), no statistics are provided. Were there any significant differences in bias measurements across the three approaches (MUA, LFP, and fMRI)? If so, this should also be indicated in Figure 1G, similar to how significant differences are reported in Figures 1C – F.

2. In Figures 2C – E, the scatter plot symbols should indicate which of the four monkeys each particular pair of bias measurements belongs to. This will make it easier for the reader to assess the consistency of the bias measurements across all fMRI runs for all monkeys. Additionally, it is unclear what constitutes a significant bias level for either of the ROIs (LGN and V1) in this fMRI dataset. Since many of the LGN/V1 bias pairings cluster around the origin (i.e., [0,0]), it would be helpful to identify significant bias pairs outside of a particular non-significant zone centered at [0,0] (shaded region: circular or oval?).

3. Throughout the results section the only statistics reported are the significance values (p-values), with the exception of the correlations reported in Table 1. The descriptive statistics (mean, standard error) should also be reported for each comparison, as well as the corresponding effect size statistic associated with the given test performed (i.e., t-statistic).

4. The correlations reported in Table 1 indicate that the correlation strength (r coefficient) between each cortical area and the LGN increases along the ventral visual pathway from V1 to V4 (mostly the case for the chromatic bias stimulus pairs, but also exists for the achromatic stimulus pair to a certain extent [V1 vs. V4]). How can this pattern of results be reconciled with the main finding that end-spectral bias is predominantly a feedforward mechanism? Given a feedforward organization one might reasonably expect the earliest cortical ROI in the feedforward chain to have the strongest correlation with the end-spectral bias it is inheriting from the LGN. Instead, the pattern of correlations appear to suggest the reverse. This warrants mention in the discussion section.

5. To what extent can the increasing trend of the homogeneity index across the ventral visual pathway (reported in Figure 3G) be attributed to fSNR differences across the ROIs? The homogeneity index as described (mean/stdev) is the inverse of how one computes the coefficient of variation (stdev/mean), with a higher CV corresponding to greater variability. Therefore, Figure 3G indicates that the chromatic

stimulus responses have the greatest variability in LGN, which decreases across the ventral visual pathway.

6. Regarding the dynamic causal modeling (DCM) results, the full set of feedback connections for each color pairing is not thoroughly tested since the V1-LGN and V2-V1 feedback connections were non-significant in the full model (all color pairings combined) and subsequently removed from each of the individual color-pairing specific models as a result. The individual color pairing DCM results would be more compelling, and provide stronger support for the feedforward basis of the end-spectral biases, if the modulatory strength of all feedback connections were estimated and compared against the corresponding feedforward connection modulatory strength.

7. No explanation is given for why the V2 node of the full DCM was not assigned a self-connection parameter.

8. Comparing the shaded error bars across Figures 5C, 5E, and 5G, it appears that the intra-laminar recordings for the achromatic trials (5G) were less noisy compared to the either chromatic bias pair (5C and 5E). Why might this be the case? Can this be explained by differences in the quality of the laminar alignment, or differences in the number/duration of recordings acquired across bias conditions?

Reviewer #3 (Remarks to the Author):

Overall:

The authors re-test the well-established empirical bias for end-spectral (relative to mid-spectral) wavelengths in 'color selective' neurons in anesthetized and awake macaques. The authors' novel goal is to use dynamic causal modeling of the color biases in different regions (including LGN, V1, V2 and V4) to infer the circuitry of this color processing feature. The authors did not attempt to clarify why such a bias exists (i.e. the teleology and/or functional utility), which simplifies the experiment, with an arguable cost to research breadth. In this version, some experimental concerns weaken the current conclusions from the dynamic causal modelling, as detailed below. Also, there is a focus on the most recent experimental information; the manuscript would be improved considerably by discussing and referring to earlier experiments in historical context; i.e. how do the current results resolve (or at least inform) prior questions about color processing?

Major Concerns:

1. By definition, 'color-selective' neurons respond selectively to variations in wavelength (e.g. positively to dominant wavelengths above 600 nm (in typical humans, termed 'red'), but negatively to wavelengths below 560 nm ("yellow-green"), or of reversed sign. Conversely, when defined strictly, such color selective neurons do not respond selectively to independent variations in luminance (the perceptual correlate of light intensity, when corrected for the sensitivity of the eye itself). Many neurons will respond to variations in both luminance and wavelength, but to varying extents. Nevertheless, to help distinguish whether a given neural response responds to variations in wavelength versus luminance, it has been standard to use flicker photometry in each subject to equate the luminance between the wavelength-varying stimuli, and to test for null responses at that "equal luminance" value (a luminance varying response), or to variations in wavelength across a range of luminance values spanning the equal luminance value. These two dimensions (wavelength and luminance) comprise two of the three independent axes of color space; spectral purity (psychophysical "saturation") comprises the third axis.

It is not clear that the authors understand these fundamental concerns and related experimental issues. For instance, in lines 500-501, the authors state: "Six equiluminous stimulus sets were used: red, green, blue, yellow, black, and white squares." By definition, variations in black and white cannot be "equiluminous"; the black and white are variations in luminance, and it is not clear that the wavelength and purity referred to as color were equal in luminance.

For instance, this manuscript does not state how "equiluminous" values were measured. If equiluminous values are measured in humans, those values do not translate exactly to those in macaque, since the macaque have a different ratio and distribution of long- to mid-wavelength sensitive cones compared to ~ 90% of humans, and the spectral sensitivity in macaques is correspondingly more protan relative to that in humans. Also, the various stimuli are described in terms of a select set of human experimenters, but those color percepts are likely different for macaques, due to these photoreceptor differences, plus differences in lens yellowing, etc. Thus the 'white' stimulus (as defined by humans) was likely 'reddish white' for the macaque.

Also, the "midspectral" colors (green and yellow) were less spectrally pure than the "red" and 'blue' (i.e. those 'colors' have a wider spectral range; and are closer to 'white' than the end-spectral colors, i.e. lower in saturation). Variations in spectral purity (or color contrast) can have a very strong influence on psychophysiological and physiological responses in 'color selective neurons.

2. Throughout the manuscript, the authors imply or state that the LGN is a beginning stage in the processing of color-opponency and/or the end-spectral bias. For instance, in the abstract, the authors state: "Our results suggest that the end-spectral bias emerges in the LGN, and then, such bias is

transmitted into V1...". Later, the authors state: "'The stronger spiking activity produced by red/blue than green/yellow was first found in the macaque primate visual cortex (Yoshioka & Dow, 1996)". Later: "...as previous studies suggest, V1 is the first place where mixing between the cone-opponent mechanism occurs (Lennie et al., 1990)".

It would be helpful if the authors could more explicitly acknowledge that color opponency arises at least as early as in the retina (in horizontal cells; see MacNichol and Svatichin, 1956) and in ganglion cells in macaque (Gouras, others). It is quite possible (perhaps likely) that the end-spectral bias arises from simple subtractive combinations of the three different photopigments in retina –though this has not been widely tested yet. In contrast, neurons in the LGN have been regarded as "relay" cells, from retina to cortex. This issue could be easily resolved by re-wording some parts of the manuscript.

3. Beyond V1, the authors model and sample activity from "V2" and higher stage areas, without taking into account that color vision processing (and the end- versus mid-spectral sensitivity differences) are (to some extent) segregated into patches which occupy only a subset of each of these areas, and have different connections with other areas), especially in the stripes in V2. Nasr and Tootell (2018, Journal of Neuroscience) discuss and measure this end- versus mid-spectral bias in human fMRI. Such color-selectivity in V2 thin stripes is also shown earlier in awake behaving macaque (Tootell et al 2004, Cerebral Cortex). Another report (Tootell et al 1988) quantifies the end- versus mid-spectral differences in different layers in macaque V1, and shows the color bias in each of the six layers of LGN (Figure 9).

4. Part of the analysis (lines 535-540) is as follows: "For each stimulus configuration, in which there were two squares, we defined two sets of sub-ROIs (one on the left hemisphere and another on the right hemisphere, corresponding to two small squares). That is, in total, there were four sub-ROIs for each color in one brain area (e.g., V1), corresponding to the four locations of squares. For each sub-ROI, based on the averaged responses across all six colors, the top 20 visual responsive voxels were selected..."

In the relevant Figure 3C, I also see four (sub) regions of interest. But given the known retinotopic organization of macaque cortex, relative to the configuration of the fixation point and the two stimuli below it (e.g. Figure 1, 2 etc.), I interpret only the two dorsalmost patches (along the V1-V2 border) of fMRI activity as driven by the stimuli of interest. The ventralmost two patches are instead located exactly at the confluent representation of the fovea, shared by V1, V2, V3. And V4, thus due to complicated visual effects on and near the foveal fixation point in the stimulus (as in most fMRI studies using a fixation task). Thus the ventralmost two of those sub-ROIs should be excluded from subsequent analysis.

Minor Concerns:

1. As I understand it, the group infers the laminar source of their signals, without direct localization such as electrical lesions or other markers of exact microelectrode tip location. Although in some cases such inferences are likely. However, tissue dimpling from the electrode array and the inherent ambiguity of localization of the extracellular sources can make such inferences less certain.

2. I do not understand this sentence; perhaps it can be re-worded: "Rather than equally encoding color information, the primary visual cortex (V1) exhibits strong bias on end-spectral colors (red and blue) relative to mid-spectral colors (green and yellow), which is similar to the black dominance found in the luminance domain."

3. The interpretation of responses to "red-green" versus "blue-yellow" (e.g. Figure 2) was challenging, because two stimulus properties (instead of only at a time) co-vary; it is not obvious whether a given difference is due to a change in yellow or blue or green or red.

4. The references for the projection of LGN (magno and parvo) to striate layers 4C (alpha and beta, respectively), e.g. lines 51-54, should include multiple primary studies by Jennifer Lund's group.

5. I don't understand the terminology used in the title and throughout this manuscript, referring to the end-spectral versus mid-spectral difference in wavelength effectiveness. Qualitatively, this effect is 'symmetrical' (on both sides of the midpoint), although there may be smaller quantitative variations in the extent of this symmetry on each side, which may also vary in different areas (is this what the authors are referring to?)

6. The authors did not cite a 'first' description of the color bias (Kruger and Gouras, 1980, J Neurophysiol). Another relevant reference is Yoshioka, Dow and Vautin, 1996.

REVIEWER COMMENTS

Reviewer #1 (Remarks to the Author):

General

The aim of the manuscript is to understand whether apparent biases for particular colors in visual cortex emerge in visual cortex, or are already present in the input to cortex. To address this the authors combine MION-enhanced fMRI measurements from LGN, V1, V2 and V4 of 4 awake macaque monkeys, with measurements of local field potential and multiunit activity (made using multichannel probes or arrays) from V1 of another 3 awake macaque monkeys.

In each case the major analyses focus on responses to a colored stimulus. This was a 4x4 degree square presented just outside the fovea, with stimuli one of red, green, blue, yellow, black, or white flashed for a couple of seconds and responses recorded. The authors find stronger LGN and V1 fMRI responses for red (vs green), blue (vs yellow) and black (vs white), and similar biases for multiunit activity and gamma LFP power in V1. The authors conclude that the color bias in V1 is largely inherited from LGN, but may be reinforced by processing in V1.

Overall, the manuscript provides a rare comparison of fMRI and electrophysiological measurements under similar viewing conditions in awake monkeys. The analyses of fMRI and electrophysiological data appear to be generally appropriate. The results are consistent with the idea that some of the biases seen in visual cortex reflect biases already generated in the retina or LGN, which is also consistent with previous work on color responses in monkey.

My major concern is that the color of the stimulus is very poorly described and it is currently impossible to reproduce the experiments, because the intensity and color of the stimuli are not provided. The methods state that the stimuli were equiluminous, but it is not clear *whether this means the colored stimuli (red, green, blue, yellow) were equiluminous to the background, or whether they were equiluminous to each other but different intensity to the background*. I assume it is the former, but this needs to be spelled out. Indeed very few details are provided: only that the electrophysiology experiments were conducted on a CRT monitor, at 45.8cd/m²; and that the fMRI experiments were conducted on a MRI-safe projector. No *details of the calibration are given (either gamma calibration of the CRT or projector primaries, or methods for color-calibration)*. At a minimum the manuscript *needs to include colorimetric definitions of the stimuli (e.g. CIE specifications)*. The authors also *need to calculate the cone-contrast of each of the stimuli, and report those cone contrasts. Without*

knowledge of the cone contrasts, some of the data are not possible to interpret. The contribution of parvocellular input layer 4Cbeta to blue bias (Fig 5E) is a case in point – is this because the blue stimulus evokes substantial responses from red or green cones, or because blue cones provide an input to the parvocellular pathway?

Reply: Thank you for the nice and important suggestions. We have added the details about calibration, equiluminance definition process, colorimetric definitions, and parameters of stimuli in the method (lines 544-560, pages 22-23; lines 632-641, page 25-26) and in supplementary materials (Supplementary Table 2, Supplementary Fig. 9).

(1) Calibration: The CRT monitor was gamma-corrected ($\gamma = 2.05$) and calibrated by LightScan software equipped with OptiCAL Photometry (Cambridge Research Systems).

(2) Equiluminance definition: In both fMRI and electrophysiology experiments, the four colored stimuli (red, green, blue, yellow) were equiluminous to the gray background (~ 10.6 cd/m² in electrophysiology experiment, ~ 5.2 cd/m² in fMRI experiment). To generate equiluminous hue stimuli and take monitor or projector gamut into consideration, we used a spectroradiometer (PR-670 SpectraScan, Photo Research Inc.) to measure the luminance of a series of gradient stimulus (red, green, blue, yellow, gray) presented by the CRT monitor and the projector separately. We then fitted the curves to find the RGB values of each stimulus (red, green, blue, yellow, gray) corresponding to the equal luminance value which is the maximum luminance that blue can achieve. In this way, we ensured all four colors and the gray background were objectively equiluminous with the chosen RGB ratio.

(3) Colorimetric definitions and parameters: The black stimuli were generated by setting the RGB value to 0, and the white stimuli had equal contrast to the background as the black stimuli. We measured the real parameters of CIE1931 color space with the spectroradiometer and further calculated the cone excitations and cone contrasts of all stimuli from CIE coordinates and luminance (Cole and Hine 1992). Following your suggestion, we provided the parameters of stimulus in multiple color spaces as Table R1 and showed the cone excitations in the MacLeod-Boynton chromaticity diagram in Figure R1.1. The parameters for each stimulus were similar to those used in previous studies (Liu et al. 2020; Valverde Salzman et al. 2012). Although there is a small value difference for stimulus luminance in fMRI and electrophysiology experiments, the cone contrasts (Table R1, Supplementary Table 2) and cone excitations (Figure R1.1, Supplementary Fig. 9) of all stimuli in the two experiments were quite similar, which is the foundation to link the imaging results with neurophysiological results.

Table R1 Parameters of stimulus in multiple color spaces in electrophysiology and fMRI experiments

	Electrophysiology				fMRI			
	Hue(°)	RGB	CIE1931 (x, y, L)	Cone Contrast (L,M,S)	Hue(°)	RGB	CIE1931 (x, y, L)	Cone Contrast (L,M,S)
Red	0	(160.7, 0, 0)	(0.61, 0.35, 10.99)	(0.32, -0.31, -0.91)	0	(176, 0, 0,)	(0.65, 0.34, 5.20)	(0.27, -0.47, -0.95)
Green	120	(0, 45, 0)	(0.29, 0.59, 10.44)	(-0.05, 0.35, -0.84)	120	(0, 130, 0)	(0.38, 0.57, 5.17)	(-0.03, 0.20, -0.89)
Blue	240	(0, 0, 255)	(0.15, 0.07, 10.02)	(-0.24, -1.80, 6.90)	240	(0, 0, 255)	(0.14, 0.09, 5.15)	(-0.23, -1.20, 9.83)
Yellow	60	(36.5, 36.5, 0)	(0.39, 0.52, 10.60)	(0.03, 0.22, -0.86)	60	(117.3, 117.3, 0)	(0.45, 0.49, 5.22)	(0.03, 0.07, -8.44)
Black	-	(0, 0, 0)	(0.24, 0.19, 0.08)	(-0.99, -1.00, -0.98)	-	(0, 0, 0)	(0.38, 0.33, 0)	(-1, -1, -1)
White	-	(61.2, 61.2, 61.2)	(0.28, 0.31, 20.40)	(0.93, 0.92, 0.96)	-	(137.7, 137.7, 137.7)	(0.35, 0.34, 10.1)	(0.96, 0.79, 1.36)
Gray	-	(30.6, 30.6, 30.6)	(0.28, 0.31, 10.55)	(0, 0, 0)	-	(117.3, 117.3, 117.3)	(0.34, 0.37, 5.22)	(0, 0, 0)

Figure R1.1 Stimuli of equal luminance in Macleod-Boynton color space

(4) As you pointed out, color contrast can have a very strong influence on the physiological responses of color-selective neurons. Based on the cone contrast (Table R1, Supplementary Table 2) and Macleod-Boynton color space (Figure R1.1, Supplementary Fig. 9), besides the difference in S cones, we could also see that blue and yellow stimuli evoked L and M cones with different relative strengths. Based on the parameters of stimulus, we speculated that the contribution of parvocellular input layer (L4Cf3) to the blue bias (Fig. 5e) is possibly because the blue stimulus evokes substantial stronger responses along L-M cone contrast axis $(-0.23 - (-1.2)) = 0.97$ than yellow stimulus does $(0.03 - 0.07) = -0.04$, which is similar to red $(0.27 - (-0.47)) = 0.74$ overtaking green $(-0.03 - 0.2) = -0.23$ alone L-M cone contrast axis. Such cone-contrast consistency between the blue-yellow pair and red-green pair was consistent with the previous study (see red arrow in Figure R1.2) (Valverde Salzman et al. 2012). As for whether S-cone signals also target L4Cf3, there is currently no clear answer of yes. One thing for sure is that all S-cone signals are relayed through the koniocellular pathway and they are not carried by the parvocellular pathway (Jayakumar, Dreher, and Vidyasagar 2013). We have added the

discussion about the explanation for parvocellular contribution to blue bias in lines 420-427 on page 17-18.

Figure R1.2 Estimated stimulus-induced L, M, S-cone contrast modulation in Valverde Salzmänn's study.

Other points

1. Line 51: At this stage it is not clear especially for a non-expert why the representation of red- and blue relative to green/yellow can be likened to the 'black dominance'. Would recommend removing.

Reply: Thank you for your suggestion. We have removed the “, which is similar to the black dominance found in the luminance domain” from the sentence (line 52-55, page3).

2. Lines 53-58: this is a long sentence and seems to imply that neurons in blobs are responsible for stronger gamma oscillations – I don't think there is evidence for that per se. Would recommend removing moving comments about blobs to discussion and restricting introduction to generic cortical circuits as in the subsequent paragraph.

Reply: Thank you for your important comment and suggestion. Here we did not intend to attribute stronger gamma oscillations to neurons in blobs and we are sorry for the ambiguity here. We have removed comments about blobs and simplified the long sentence into two short sentences for easier understanding (lines 59-64, page 3).

3. Line 60: There are a few things going on in this paragraph. I think generally that what you are saying is simply that bias for particular colors may arise because the input to cortex has that bias, or because of the way that cortex its inputs. However, this is muddled somewhat because many of the comments about inhibition and recurrence either need to be spelled out more or omitted. It's not clear to me that

they add to the main point here – could be better off simply saying that you want to establish this, by characterising the input, and the cortical processing, as you do in the final paragraph.

Reply: Thank you for your suggestion. We have simplified the expressions by omitting the terms “recurrence” and “inhibition” in this paragraph (lines 66-75, pages 3-4). We used “cortical mechanism” instead.

4. Line 67: not sure what you mean by ‘high density’ of neurons preferring red- and blue. Do you simply mean that there are more of them in the superficial layers than neurons preferring other colors?

Reply: Yes, your interpretation is correct. We use ‘high density’ to illustrate the neurons preferring red or blue outnumbered neurons preferring yellow or green. For easier understanding, we removed ‘high density’ and changed the sentence to “... considering that there is larger segregated clustering of neurons preferring red and blue than blue and yellow” in lines 71-73 on page 3.

5. Line 89: you say “one monkey” for linear probes but note both DS and QQ used here and in methods. You then say “the three monkeys were trained” on line 92 but note 4 animals in Figure 1 legend. This is all a bit confusing – please clarify.

Reply: We apologize for the confusion. We used two monkeys (DS and QQ) for laminar recording by linear probes. And four monkeys in total were trained for electrophysiology experiments. We have clarified the number in the revised Results (lines 96-97, page 5).

6. Line 93: In fMRI methods on line 506 you state a 2s stimulus epoch, here it is 3s. Please clarify (and make in clear in both electrophysiology and fMRI methods sections).

Reply: We apologize for the confusion. We have changed the 3s of the stimulus epoch to 2s in line 99 on page 5. We also made it more clear in the revised Methods (lines 569, 571-572, page 23).

7. Line 94: You say that the in some cases you made the stimulus ‘cover half RF by stimulus edges’. I am not sure I understand this – an image showing the RFs of the neurons, and the outline of the stimulus, would be useful.

Reply: Thank you for your nice suggestion. We added a simplified illustration of RF locations in Figure 1a and provided the RF distribution and stimulus positions in Figure R1.3 and Supplementary Fig. 1.

We also rephrase the sentence as “In each trial, the center or edge of the square was shown in the RFs of the recorded sites (Fig. 1a)” in line 101 on page5.

Figure R1.3 Illustration of stimulus presentation and RF distribution.

a. RF centers and sizes from one array of monkey DQ. Red squares represent two of five possible stimulus positions (left, right, top, down edge of a square in the RFs, surface center in the RFs). **b.** 5 relative positions of RF to the stimulus outline. **c.** Simplified illustration of stimulus presentation and RF locations relative to the stimulus.

8. Line 95: I am not sure what you mean by ‘ensured that the neural response to the equal-luminance stimulus was lowest’). Please clarify.

Reply: Thank you for your comment. The statement ‘**We also ensured that the neural response to the equal-luminance stimuli was lowest by conducting a control experiment with varying hues and luminance**’ has been removed from line 102 on page 5. In the last version of the manuscript, the stimulus luminance we used in the probe experiment was defined by the tuning results in a luminance-varying experiment. In this experiment, we recorded the neural response from V1 layer 4Ca and 4B (M-pathway specific) driven by a series of hue stimuli with multiple levels of luminance (relative to the luminance of gray background: 10.6 cd/m²). We chose the luminance value that corresponds to the lowest neural activation (dashed vertical line in Figure R1.4, Supplementary Fig.6) in the formal experiment to avoid the contamination of luminance-selective neurons.

Based on the suggestions of reviewer #3 (Q1), we have performed a new probe experiment by using the same stimuli as in the array experiment (i.e. hues physically equiluminant to gray background) and used the new probe result in the main text of the manuscript (Figure 1&5). Therefore, the statement has been removed from line 102. And the results of luminance-matching experiments and corresponding laminar experiments were shown as control experiments in the supplementary materials (Supplementary figure 6-7) and they were described in the main text (lines 337-349, page 15).

Figure R1.4 Response from V1 layers 4C α and 4B in the luminance-matching experiment.

In the luminance-matching experiment, we used a fast square-wave drifting grating (SF = 2 Hz, temporal frequency = 16.67 Hz, preferred orientation) composed of background gray (10.6 cd/m²) and red/green/blue/yellow with multiple levels of luminance. Dashed vertical black lines indicate the RGB values in the x-axis corresponding to the minimal neural response, which is used in the control laminar electrophysiology experiment.

9. Line 156: I am not sure what the correlations in Fig 2C-E imply. I think that each point represents one run from one animal, but its not clear to me why individual runs are been shown here. The unit of observation would seem to be the animal, so analysis that averaged runs within animals, or somehow took account of animal as a grouping factor would seem to be important to be able to show that bias within LGN predicts bias within V1, or vice versa.

Reply: We thank the reviewer for pointing this issue out. Given the sample size (n=4) in the present study, we treated each run as one sample in the correlation analyses. We agree with the reviewer that incorporating the subject as a grouping factor would enhance the rigor of our analyses. To address this concern, we performed partial correlation analyses, including the subject as a control variable, by converting it into binary values (Hayes and Preacher 2014). We were still able to observe significant correlations between LGN and V1 in the context of red-green bias ($r = 0.219$, $p = 0.030$), blue-yellow bias ($r = 0.293$, $p = 0.003$), and black-white bias ($r = 0.389$, $p < 0.001$). We have updated all the correlation analyses in the revised manuscript besides these (e.g., Supplementary Table 1) and found similar results to our initial ones. Moreover, we incorporated the comment of reviewer #2 Q#2.2 and labeled the data from each subject with different symbols (please see below and Figure 2c-e). We have

updated this information to the revised Methods and Results (line 180-182, page 8; line 761-763, page 30).

Figure R1.5 Correlation between LGN and V1 for bias indices.

a-c. Correlations between LGN and V1 for the red-green bias (a), blue-yellow bias (b), and black-white bias (c). Each monkey was labeled with different symbols. Shadow region represents 95% confidence intervals.

10. Line 177 and around it: The analyses suggest a difference between V1 and V4 (and less, if any difference between V1 and V2, or V2 and V4) in color bias. Comparisons of responses in V1, V2 and V4 are made difficult by the fact that the retinotopy in the three regions is quite different. The ROIs appear to be based on the voxels that were responsive to each of the two visual stimulus locations, introducing possible circularity. I would like to see some analysis of voxel response that took into account the possibility that color bias depends on eccentricity, and asked whether changes in V4 survive such analysis.

Reply: We appreciate the reviewer's insightful comment.

Firstly, we apologize for any confusion caused by the lack of clarity in our ROI definition. When defining ROIs, we used the average response across all six stimuli (i.e., red, green, blue, yellow, black, and white). It is worth mentioning that only the paired two stimuli were used in calculating the bias index (i.e., red-green bias, blue-yellow bias, and black-white bias). Notably, the distribution of the six conditions was uniform throughout the experiment. In this case, the impact of circularity on our findings should be limited based on discussions on circularity in the previous study (Kriegeskorte et al. 2009). To further address the issue of circularity and to provide reassurance regarding the robustness of our results, we conducted an additional analysis using a stricter approach. We split the data into two sets: odd and even runs. We defined the ROIs using only one-half of the data (e.g., odd) and examined

the responses on the other half (e.g., even). Then, we repeated this process in reverse, using the even runs to define ROIs and examining the responses in the odd runs. Finally, we averaged the results obtained from both steps. The results from this analysis demonstrated a similar trend to our initial findings, albeit with slightly reduced significance due to the reduced sample size (Figure R1.6 and Supplementary Fig.10).

Figure R1.6 fMRI responses to each stimulus, bias indices, and homogeneity indices among the LGN, V1, V2, and V4 when defining ROIs using one-half data (odd or even runs) and testing on the other half.

a-d. Averaged fMRI responses to three color pairs across all four subjects in the LGN (a), V1 (b), V2 (c), and V4 (d). * $p < 0.05$, ** $p < 0.01$, *** $p < 0.001$ in comparisons of the generalized linear mixed model (GLMM) analyses. **e-g.** The change trends of red-green bias (e), blue-yellow bias (f), and black-white bias (g) from LGN to V4. * $p < 0.05$, ** $p < 0.01$, *** $p < 0.001$, post hoc comparisons of GLMM with Bonferroni corrections. **h.** Color homogeneity across visual areas. ** $p < 0.01$, *** $p < 0.001$, post hoc comparisons of GLMM with Bonferroni corrections. Error bars indicate standard error. ROIs were defined using one-half data (odd or even runs) and responses were checked on the other half.

Indeed, the retinotopic distribution differs among V1, V2, and V4. In our experiments, the monkeys were required to maintain fixation on a square in the center of the screen. When fixation was maintained well [mean (SE): 88.08% (0.86%) in the present study], corresponding receptive fields

(around 2-6 degrees in eccentricity) covered by the stimulus were mainly activated. Therefore, the current findings may mainly hold for the areas/voxels within receptive fields ranging from 2-6 degrees. The end-spectral bias outside this range could not be accessed in the present study.

It would be ideal to define the retinotopic map for each monkey and precisely assess the distribution of bias based on eccentricity to address the reviewer's concern about the potential impact of eccentricity on the current findings. However, due to the specific characteristics (e.g., rapidly movable) of the stimuli required for localizing the retinotopic map, extensive training is necessary to acquire a reliable retinotopic map, which poses challenges to the health of animals with implanted head posts and the potential signal sensitivity reduction due to the accumulation of MION contrast agents. After several unsuccessful attempts, we proceeded without localizing the retinotopic map for each monkey. Instead, we attempted to find a shared template of the monkey retinotopic map as is typically done in human studies. Unfortunately, we were unable to find such a shared template. Therefore, to address the reviewer's concern more comprehensively, we performed the following analyses.

We aligned our data with the datasets shared by Janssens et al., (2014), which provides the probabilistic map of V1, V2, V3, and V4. Based on these probabilistic maps, we marked the approximate location of the fovea (indicated by a black star in Figure R1.7). We then examined the distribution of the bias index within each ROI (see Figure R1.7a for an example from monkey Q), comparing it with the retinotopic map shown in previous studies (e.g., Arcaro and Livingstone 2017). No prominent patterns (e.g., a hub-and-spoke distribution from the fovea) were observed (Figure R1.7a).

Figure R1.7 The distribution of the bias indices in defined ROIs and the retinotopic map.

a. Three kinds of bias indices (from left to right, red-green bias, blue-yellow bias, black-white bias) for each voxel within each ROI of monkey Q. Black star marks the approximate location of the fovea. The borders of V1, V2, and V4 are indicated by the black dashed lines. **b.** The retinotopic map from Arcaro and Livingstone, 2017.

Next, to conduct a quantitative analysis on potential biases among voxels with varying eccentricities within each ROI, we divided each ROI into two sub-ROIs, one closer to and the other farther from the fovea, according to previous studies (Janssens et al. 2014; Arcaro and Livingstone 2017). No significant differences were found between these two sub-ROIs in the three bias indices (see Figure R1.8).

Additionally, we performed bias calculations and statistics within the defined ROIs at the voxel level. We treated each voxel instead of each run as a sample. Our results showed that the changes in bias from LGN to V4 still existed (Figure R1.9).

The above findings suggested the bias of individual voxels within each ROI and the changing trend from LGN to V4 remained relatively stable within the receptive field range of 2-6 degrees.

Figure R1.8 Bias indices in sub-ROIs closer to and farther from the fovea.

a-c. Red-green bias (a), blue-yellow bias (b), and black-white bias (c) in V1, V2, and V4 when splitting the voxels in each ROI into two parts: one close to the fovea and the other away from the fovea. ns, not significant in comparisons of GLMM analysis. **d.** Schematic diagram of the two sub-ROIs. The sub-ROI closer to the fovea is marked in orange, and the farther part is marked in red. Black star marks the approximate location of the fovea. The borders of V1, V2, and V4 are indicated by the black dashed lines.

Figure R1.9 The change trend of bias indices from LGN to V4 at the voxel level.

a-c. The change trends of red-green bias (a), blue-yellow bias (b), and black-white bias (c) from LGN to V4. Each point represented one voxel from one subject. * $p < 0.05$, *** $p < 0.001$ in post hoc comparisons of GLMM with Bonferroni corrections. Error bars indicate standard error.

Though no studies directly investigated the end-spectral bias across subregions with different eccentricities in V1, V2, and V4, previous evidence has reported no variation in color sensitivity (at least between 0° and 10°) in V1 (Tootell et al. 1988). Notably, a previous study conducted in macaques by Liu et al. (2020) demonstrated that when using full-screen stimuli and recording signals in regions corresponding to the peripheral eccentricity at V1, V2, and V4, a gradual homogeneity among different hues from V1 to V4 was observed, consistent with our findings within the receptive field range of 2-6 degrees. Therefore, the changes in the end-spectral bias from V1 to V4 might exhibit consistency across different eccentricities even outside the 2-6 degrees in the present study.

Taken together, the changing trend in the end-spectral bias from LGN to V4 and the end-spectral bias within these regions might remain relatively stable within the receptive field range of 2-6 degrees in eccentricity and even outside this range, which might be interesting to investigate in future studies.

In the revised manuscript, we have discussed the potential impact of eccentricity on the current findings as below (lines 485-498, page 20).

11. Lines 190-193. I don't actually understand this part. Could you find other ways to state this to make it easier?

Reply: We apologize for not making the description of this section clear. Our objective was to investigate potential changes in differences among four colors along the visual pathway. To quantify these changes, we calculated the homogeneity index as the average of the responses to each of the four colors, divided by their variance (Eq.6). A higher homogeneity index indicated less variation among the four colors. As depicted in Figure 3g, our findings indicated a consistent increase in the homogeneity index from LGN to V4, suggesting a rise in uniformity of color representation. As shown in Figure R1.10 and Figure S4, responses from V1 to V4 gradually converged across the four colors, with red and blue exhibiting closer responses and yellow and green as well. Moreover, the reduction of red-green bias might also contribute to the uniformity of color representation V1 to V4 (Fig. 3d). We have added this information to the revised manuscript (lines 228-239, page 10).

Figure R1.10 The fMRI stimuli in the four ROIs.

Each stimulus is represented by its corresponding color. Error bars indicate standard error.

12. Line 231: Is it possible that the greater feedforward strengths for red, blue, black simply reflect stronger responses to those colors? It would be useful to distinguish.

Reply: Indeed, the magnitude of the response can influence the evaluation of connectivity strength in DCM analyses. To account for this, following the approach outlined in the previous guideline (Zeidman et al. 2019), we performed a principal component analysis (PCA) before conducting the DCM analyses. This preprocessing step involved extracting the principal eigenvariate, which effectively captured the data characteristics beyond the average amplitude changes typically reflected by the mean value. By doing so, the impact of response variations across experimental conditions was mitigated. As a result, when a greater feedforward connection in our results was observed, it primarily reflected a stronger modulation of effective connectivity due to the experimental conditions rather than indicating stronger responses within the individual nodes themselves. We have added this information to the revised Methods (lines 730-734, page 29).

13. Figure 5: It would be useful to see some examples of the responses that go into making the line graphs in panels C, E, & G.

Reply: Thank you for your suggestions. We have added examples of the responses from single sites in Figure 5 (also shown in Figure R1.11 below). The six insets showed the dynamic MUA responses from two example sites at output and input layers, which were simultaneously recorded in the same probe placement.

Figure R1.11 Laminar pattern of response bias in V1

Left panel, laminar pattern of red–green bias in V1. Data are shown as the mean values with standard error. The two insets show the dynamic MUA responses from two example sites at the output and input layers, which were simultaneously recorded within the same probe placement. Medium and right panels, the laminar pattern of blue-yellow bias and black-white bias similar to the left panel.

14. Line 323: many other groups have reported asymmetries of various sorts in LGN and in V1.
 Reply: Thank you for pointing out the previous findings. We have added the references of the reports on other kinds of color and luminance asymmetries in LGN and in V1 (Mullen, Dumoulin, and Hess 2008; Solomon and Lennie 2005; Kremkow et al. 2014) and corresponding discussion in lines 402406 on page 17. And we have removed the ‘for the first time’ from the sentence in line 402 on page 17.

Solomon and Lennie found (2005) the asymmetry of normalization strength between +S cells (cells receiving excitatory S-cone input) and L-M cells (parvocellular cells) in LGN. Mullen et al found (2008) the asymmetry of contrast sensitivity of red-green and blue-yellow gratings in LGN. The asymmetries of other sorts in LGN together with the current finding of end-spectral bias in LGN commonly suggest a pre-cortical substrate for the unbalanced color representation. Beyond LGN, we further uncovered the increasing trend of blue-yellow bias from LGN to V4, which suggest an amplification of the S-cone signal during cortical processing (Mullen, Dumoulin, & Hess, 2008).

15. Line 402: This seems to imply recordings from 3 animals, but include 4 – reword to make clearer the total number of animals.

Reply: Thank you for pointing out the mistake. We have changed 3 to 4 in line 525 on page 22.

16. Line 412: 'After the signal dying out' – I understand what you mean, but this should be made clearer.

Reply: Thank you for your suggestion. We have changed 'dying out' to 'decayed with time' in line 534 on page 22.

Reviewer #2 (Remarks to the Author):

General Comments

This manuscript employs multiple data acquisition techniques to thoroughly demonstrate the prevalence of the end-spectral bias throughout the ventral visual pathway in non-human primates, ultimately building a coherent case for a subcortical origin and a feedforward progression of this well-studied neural coding privilege. The strongest evidence being the laminar response pattern in V1, with peaks predominantly found in the granular input layer, and the consistent color pairing response differences (end-spectral bias) across both invasive and non-invasive recording techniques. The substantially weaker (R-G, B-W) or nonsignificant (B-Y) end-spectral bias findings in LGN compared to the cortical ROIs, the potential for fSNR differences across ROIs, and the stronger LGN bias correlations with extrastriate visual areas compared to the striate visual area (V1) adds some uncertainty to the feedforward conclusions being drawn.

More broadly, this study strongly reinforces the link between neurophysiology and functional neuroimaging (fMRI) techniques by demonstrating qualitatively similar results across these experimental domains while using an identical behavioral paradigm. Additionally, it provides a straightforward framework for reproducing and building upon the reported findings further with human neuroimaging experiments.

There are a few minor points related to the methods which require further clarification, but in general the methods provide adequate detail for the work to be understood and reproduced. There are several concerns related to how certain results are reported and interpreted which requires some further explanation and revision in order to fully support the main conclusions of the study.

Reply: We appreciate your positive comments and recognition for our study. Following your valuable suggestions, we have provided more details related to the methods and have added more results from control analysis to help to enhance explanatory persuasiveness. Thank you very much!

Methods

1.1. There are inconsistencies in the reported number of monkeys used in the electrophysiology experiments. At multiple points in the results and methods sections it is mentioned that three monkeys were used for data collection, but subsequently the IDs for 4 monkeys are reported.

Reply: thank you for pointing out the discrepancy on numbers. We used four monkeys in total in electrophysiology experiments and we changed the ‘three’ in Results (line 97, page 5) and Methods (line 525, page 22) to ‘Four’.

1.2. When describing the RF mapping procedure, the (Michelson?) contrast level of the sparse noise stimulus is reported in units of degrees, instead of percentage. Furthermore, it is unclear if the “0.1° - 0.3°” actually refers to the range of visual angle which the sparse noise stimulus occupied, or is somehow related to the contrast level.

Reply: Thank you for pointing out the typo. We mistakenly added a ‘°’ after the ‘contrast 0.9’ which corresponding to 90%. We have removed the ‘°’ in line 585 on page 24. The “°” represent the degree for visual angle.

1.3. Within the fMRI data analysis section, it states that “All fMRI signals throughout the study were inverted so that an increase in signal intensity indicates an increase in activation”. Further explanation and justification for this uncommon fMRI preprocessing step should be provided as it is unclear why it was necessary. Is this inversion necessary when using the MION contrast agent?

Reply: Thank you for bringing up this point. We apologize for not clearly explaining this step in our methods. In macaque experiments utilizing MION (monocrystalline iron oxide nanocolloid) as the contrast agent, upon activation, the fMRI signals undergo a decrease due to local iron concentration increasing instead of the typical increase observed in BOLD signals (Figure R2.1 from Leite et al. 2002, MION: red line, BOLD: black line).

To facilitate the comparison of our results with those commonly reported in human studies, where the BOLD signal is typically presented, we performed an inversion of all fMRI signal changes throughout our study. This inversion allows for a more intuitive interpretation, where an increase in signal intensity now corresponds to an increase in activation.

Figure R2.1 The comparison of BOLD and MION.

Relaxation rate changes for BOLD (black) and MION (red) contrast, together with the corresponding linear model fits (from Leite et al., 2002).

Results

2.1. When reporting the results presented in Figure 1G (last paragraph before the “End-spectral and polarity bias in LGN” section), no statistics are provided. Were there any significant differences in bias measurements across the three approaches (MUA, LFP, and fMRI)? If so, this should also be indicated in Figure 1G, similar to how significant differences are reported in Figures 1C – F.

Reply: Thank you for your suggestion. We have added the results of inference statistics in line 148154 on page 6-7 and in Figure 1g (also shown in Figure R2.2 below). For end-spectral biases (red-green, blue-yellow), We found a significantly larger bias index calculated from the gamma power of LFP than those calculated from MUA ($p < 10^{-10}$) or BOLD signals ($p < 10^{-9}$ in post hoc comparisons with Bonferroni corrections of one-way ANOVA, red-green bias: $F_{2,1182}=70.99$, $p < 10^{-10}$, blue-yellow bias: $F_{2,1182}=99.59$, $p < 10^{-10}$). For luminance bias, we found stronger black dominance based on MUA than LFP ($p < 10^{-10}$) or BOLD signals ($p < 10^{-10}$ in post hoc comparisons with Bonferroni corrections of one-way ANOVA, $F_{2,1182}=108.63$, $p < 10^{-10}$). We did not find any significant difference in red-green or blue-yellow bias between MUA and fMRI approaches ($ps = 1$, post hoc comparisons with Bonferroni corrections). These nonsignificant comparison results between the two measurements (neurophysiology and fMRI) became a foundation for us to link the classic phenomenon, end-spectral bias, from the two fields and further investigate its source and cortical transition.

Figure R2.2 Color and luminance biases detected by three approaches.

*** $p < 0.001$ in post hoc comparisons with Bonferroni corrections of one-way ANOVA. Error bars indicate standard error.

2.2. In Figures 2C – E, the scatter plot symbols should indicate which of the four monkeys each particular pair of bias measurements belongs to. This will make it easier for the reader to assess the consistency of the bias measurements across all fMRI runs for all monkeys. Additionally, it is unclear what constitutes a significant bias level for either of the ROIs (LGN and V1) in this fMRI dataset. Since many of the LGN/V1 bias pairings cluster around the origin (i.e., [0,0]), it would be helpful to identify significant bias pairs outside of a particular non-significant zone centered at [0,0] (shaded region: circular or oval?).

Reply: We are grateful to the reviewer for this valuable comment. As suggested by the reviewer, we have made the following improvements to the analysis and visualization of bias pairs.

Firstly, we labeled the bias pairs for each monkey with different symbols (Figure R1.5). Additionally, we have marked the 95% confidence interval of linear regression using shaded regions (Figure R1.5 and Figure 2c-e).

To address the concern of potential bias introduced by extreme bias pairs, we implemented a filtering approach. We set the third quartile of the maximum bias index value (1) as a threshold. The bias pair, whose absolute bias index value surpassed the threshold (0.75) in either the x or y dimension, was identified as further away from the origin and marked with a gray color. Subsequently, we removed these marked bias pairs and conducted the correlation analysis again. Notably, even after removing the bias pairs with extreme bias indices, the correlations between LGN and V1 remained significant (Figure R2.3 & Figure S2b, red-green bias: $r = 0.255$, $p = 0.029$; blue-yellow bias: $r = 0.267$, $p = 0.012$; black-white bias: $r = 0.228$, $p = 0.036$). This finding suggested that the relationship between

LGN and V1 was robust and not solely driven by individual bias pairs with extreme bias indices. We have added this information to the supplementary materials.

Notably, as suggested by Reviewer #1 Q#9, we have incorporated the grouping factor of the monkey as a control variable in the revised correlation analyses. This adjustment also helps to account for potential inter-individual differences among monkeys and strengthens the validity of our analysis.

Figure R2.3 Correlation between LGN and V1 for bias indices after removal of extreme values.

a-c. Correlations between LGN and V1 for the red-green bias (a), blue-yellow bias (b), and black-white bias (c) after removing data further away from the origin. Extreme bias indices (absolute value surpassed the threshold: 0.75) are marked with gray color. Each monkey is indicated by a different symbol. The shaded region represents 95% confidence intervals.

2.3. Throughout the results section the only statistics reported are the significance values (p-values), with the exception of the correlations reported in Table 1. The descriptive statistics (mean, standard error) should also be reported for each comparison, as well as the corresponding effect size statistic associated with the given test performed (i.e., t-statistic).

Reply: Thank you for your suggestion. We have provided all the descriptive statistics and inference statistics for each comparison.

2.4. The correlations reported in Table 1 indicate that the correlation strength (r coefficient) between each cortical area and the LGN increases along the ventral visual pathway from V1 to V4 (mostly the case for the chromatic bias stimulus pairs, but also exists for the achromatic stimulus pair to a certain extent [V1 vs. V4]). How can this pattern of results be reconciled with the main finding that end-spectral bias is predominantly a feedforward mechanism? Given a feedforward organization one might reasonably expect the earliest cortical ROI in the feedforward chain to have the strongest correlation

with the end-spectral bias it is inheriting from the LGN. Instead, the pattern of correlations appear to suggest the reverse. This warrants mention in the discussion section.

Reply: We greatly appreciate the reviewer for pointing this issue out. According to anatomical connections of the primate visual system (i.e., the full model used in the DCM analysis), the presence of a direct connection between LGN and V4 is questionable (Girard, Salin, and Bullier 1991). Therefore, we speculated that V1 and V2, which directly connect with LGN and/or V4, might be mediators in the correlation observed between LGN and V4. Indeed, when we set V1 and/or V2 as control variables, the correlation strength between LGN and V4 was reduced (Table R2). We acknowledge the presence of mediation effects in the correlation analysis, which can complicate the interpretation of the underlying source of the correlation. Therefore, we refrained from solely relying on the correlation analysis to explain the relationship among these regions. Instead, the correlation analysis between the LGN and V1 primarily served as a preliminary indication for further DCM and laminar activity analysis. We have made this point clear in the revised manuscript. Moreover, to limit the potential confusion about these correlation results, we have added the above-mentioned partial correlation results to the supplementary materials (Supplementary Table 1) and discussed the correlation results in the revised manuscript (line 209-213, page 9):

Table R2 The correlation strength between LGN and V4 when controlling for V1 and/or V2.

Controlled variables		r(p)		
		Red-Green bias	Blue-Yellow bias	Black-White bias
LGN-V4	-	0.330 (p<0.001)***	0.439 (p<0.001)***	0.509 (p<0.001)***
LGN-V4	V1	0.260 (p=0.010)*	0.355 (p<0.001)***	0.385 (p<0.001)***
LGN-V4	V2	0.175 (p=0.087)	0.326 (p=0.001)**	0.393 (p<0.001)***
LGN-V4	V1, V2	0.168 (p=0.101)	0.290 (p=0.004)	0.352 (p<0.001)***

* $p < 0.05$, ** $p < 0.01$, *** $p < 0.001$.

2.5. To what extent can the increasing trend of the homogeneity index across the ventral visual pathway (reported in Figure 3G) be attributed to fSNR differences across the ROIs? The homogeneity index as described (mean/stdev) is the inverse of how one computes the coefficient of variation (stdev/mean), with a higher CV corresponding to greater variability. Therefore, Figure 3G indicates that the

chromatic stimulus responses have the greatest variability in LGN, which decreases across the ventral visual pathway.

Reply: We thank the reviewer for bringing this issue to our attention. We agree with the reviewer that the SNR may impact the homogeneity index. In the present study, the homogeneity index was calculated as the average of the responses (A) to each of the four colors, divided by their variance (σ_A) (Eq.R1). It should be noted that the SNR is typically computed as the ratio of the mean signal intensity (S) to the standard deviation of the noise (σ_N) (Eq.R2)

$$\text{Homogeneity index} = \frac{A(4 \text{ stimuli})}{\sigma_{A(4 \text{ stimuli})}} \quad \text{Eq. R1}$$

$$\text{SNR} = \frac{S}{\sigma_N} \quad \text{Eq. R2}$$

$$\text{CNR} = \frac{A(6 \text{ stimuli})}{\sigma_N} \quad \text{Eq. R3}$$

We evaluated the SNR in the four regions (Figure R2.4a) and observed a similar trend in the SNR as that of the homogeneity index from V1 to V4 but with a higher SNR in LGN. Previous studies have demonstrated that MION causes signal intensity decreases (Leite et al. 2002). As such, the higher SNR in LGN might be due to fewer decreases after the MION injection or higher SNR in LGN even without MION. Nevertheless, to further address whether the SNR impacted the homogeneity index, we treated the SNR as a covariate and reanalyzed the data. The new analyses yielded results consistent with the initial findings (Figure R2.4c), suggesting that the trends in the homogeneity index might not solely be attributed to the SNR. Note that the contrast-to-noise ratio (CNR, calculated as responses (A) divided by the noise variance (σ_N), Eq.R3) increases after MION injection. Moreover, CNR could better measure the activation fluctuations relative to the noise in fMRI (Welvaert and Rosseel 2013). Therefore, we compared the CNR in the four regions (Figure R2.4b). We found a similar trend in the CNR as that of the homogeneity index from LGN to V4. Indeed, the CNR in LGN was significantly lower than that in other regions. Again, we treated the CNR as a covariate and reanalyzed the data. The new analyses yielded results consistent with our initial findings (Figure R2.4d), suggesting that the trends in the homogeneity index might not solely be attributed to the CNR. Taken together, in LGN, the relatively lower homogeneity index might be attributed to the greater variability, as the reviewer pointed out, while the increased homogeneity index along the ventral visual pathway may be due to the decreased differences in responses evoked by the four colors. As shown in Figure R1.10, responses from V1 to V4 gradually converged across the four colors, with red and blue exhibiting closer responses and yellow and green as well. We have added the above-mentioned CNR results in the revised manuscript (line 234-236, page 10; line 717-722, page 28)

Figure R2.4 The signal-to-noise ratio (SNR), contrast-to-noise ratio (CNR) of the LGN, V1, V2, and V4 and the change in the homogeneity index when SNR or CNR is treated as a covariate.

a-b. The Signal-to-Noise Ratio (SNR) and Contrast-to-Noise Ratio (CNR) of LGN, V1, V2, and V4. **c-d.** Differences in the homogeneity index among four regions after SNR and CNR were treated as covariates, respectively. * $p < 0.05$, ** $p < 0.01$, *** $p < 0.001$, post hoc comparisons of GLMM with Bonferroni corrections. Error bars indicate standard error.

2.6. Regarding the dynamic causal modeling (DCM) results, the full set of feedback connections for each color pairing is not thoroughly tested since the V1-LGN and V2-V1 feedback connections were non-significant in the full model (all color pairings combined) and subsequently removed from each of the individual color-pairing specific models as a result. The individual color pairing DCM results would be more compelling, and provide stronger support for the feedforward basis of the end-spectral biases, if the modulatory strength of all feedback connections were estimated and compared against the corresponding feedforward connection modulatory strength.

Reply: Thanks for your suggestion. We apologize for any confusion caused by the lack of clarity in the DCM section. Since all the color pairs were presented in each run with a random order, especially given a rapid-event fMRI design utilized in the present study, the responses evoked by the preceding stimuli might persist, albeit weak, when the subsequent event was presented. Therefore, only fitting the data with partial conditions in the DCM analyses might cause improper estimation. Therefore, we

evaluated Matrix B, which represents the modulatory or exogenous input, with all the stimuli. Note that all the connections in the full model were set for Matrix B. That is, although the feedback connections of V1-LGN and V2-V1 were not significant in Matrix A (representing intrinsic or endogenous connectivity), the modulatory effects evoked by stimuli on these connections were still fitted in Matrix B.

In the initial submission, our comparison focused on the modulation differences in the significant connections identified in Matrix A. To address the concern raised by the reviewer, we showed the results of Matrix B on all the feedforward and feedback connections in Figure R2.5 a-c & Figure S5, in which the insignificant feedback connections in Matrix A (i.e., V1-LGN and V2-V1) were depicted with unfilled bars. As shown in Figure R2.5, the differences in modulatory effects between red and green, blue and yellow, as well as between black and white on the V1-LGN and V2-V1 feedback connections, were minor or negative. Notably, the modulatory effects on these connections primarily exhibited inhibitory. As such, the negative differences indicated a stronger inhibitory feedback modulation effect evoked by red, blue, and black than green, yellow, and white, respectively. These results provided stronger support for the feedforward basis of the end-spectral biases. We have added a description of this part in the main text (line 289-292, page 13).

Figure R2.5. Differences between all feedforward and feedback modulatory connections in three pairs.

a-c. Differences between modulatory connections evoked by the two stimuli in red-green (a), blue-yellow (b), and black-white (c) pairs. ** $p < 0.01$, *** $p < 0.001$ in comparison of GLMM analysis. Error bars indicate standard error.

2.7. No explanation is given for why the V2 node of the full DCM was not assigned a self-connection parameter.

Reply: We appreciate the reviewer for bringing up this point. We did include the self-connection of V2 in the full model. Upon evaluation, we found that the posterior probability of the self-connection of V2 was 0.810, which did not meet the threshold that we set at 0.95. As such, the self-connection of V2 should be marked as a dotted line in Figure 4. We apologize for this oversight. We have corrected it in the revised figure (Figure 4).

2.8. Comparing the shaded error bars across Figures 5C, 5E, and 5G, it appears that the intra-laminar recordings for the achromatic trials (5G) were less noisy compared to the either chromatic bias pair (5C and 5E). Why might this be the case? Can this be explained by differences in the quality of the laminar alignment, or differences in the number/duration of recordings acquired across bias conditions?

Reply: This is an interesting question. During daily recording sessions, all six stimuli (red, green, blue, yellow, black, and white) were randomly mixed in each block. Therefore, the difference in noise level between achromatic and chromatic biases was not due to the distinct laminar alignment or number/duration of recordings for each stimulus condition.

We speculated that the higher noise level for the chromatic bias pair than the achromatic bias pair might reflect the larger variation of chromatic preference than achromatic preference in the probe locations recorded in this experiment. In the new control experiment of laminar recording, we used the same black and white stimuli together with four color stimuli (red, green, blue, yellow) equiluminant to the background, and we found the noise levels of chromatic pair and achromatic pair were roughly the same.

Figure R3.3 Laminar pattern of end-spectral bias and black dominance by using two sets of stimuli.

a. Laminar distribution of bias indices with stimulus luminance matched by M pathway. **b.** Laminar distribution of bias indices of with physically equiluminous stimulus (same as Figure 5 in the manuscript).

Reviewer #3 (Remarks to the Author):

Overall:

The authors re-test the well-established empirical bias for end-spectral (relative to mid-spectral) wavelengths in ‘color selective’ neurons in anesthetized and awake macaques. The authors’ novel goal is to use dynamic causal modeling of the color biases in different regions (including LGN, V1, V2 and V4) to infer the circuitry of this color processing feature. The authors did not attempt to clarify why such a bias exists (i.e. the teleology and/or functional utility), which simplifies the experiment, with an arguable cost to research breadth. In this version, some experimental concerns weaken the current conclusions from the dynamic causal modelling, as detailed below. Also, there is a focus on the most recent experimental information; the manuscript would be improved considerably by discussing and referring to earlier experiments in historical context; i.e. how do the current results resolve (or at least inform) prior questions about color processing?

Reply: Thank you for your comments. In the current study, we try to investigate 1) the contribution of feedforward pathways (especially P and K pathways) and 2) the cortical processing for the well-established bias for end-spectral wavelengths (relative to mid-spectral) in the visual cortex of awake macaques, by combining the laminar recording and fMRI technology. As you mentioned, we did not intend to illustrate why such a bias exists and we hope the following new experiments, analysis, and discussions may further strengthen the research novelty.

According to your systematic instructions, we did a new laminar experiment to illustrate the laminar consistency of end-spectral bias between two sets of stimuli (with physically equal luminance and with equal luminance measured by minimizing the neural response in the M pathway). Also, we provided additional control analysis for color patches to make our conclusions more solid. Finally, we also cited more earlier studies (e.g. Tootell et al 1988) to discuss the current results in historical context. We found that the LGN and V1 laminar pattern of end-spectral bias in awake monkeys was highly consistent with Tootell’s study (1988), which suggests a dominant contribution of the P pathway for long- and short-wavelength bias. Furthermore, we also found that LGN is not the only contributor to the end-spectral bias, LGN-V1 projections to L4Cb also enhance such bias, and downstream cortical processing further weakens the response to long wavelength and strengthens the response to short wavelength relative to mid-spectral wavelengths.

Major Concerns:

1. By definition, 'color-selective' neurons respond selectively to variations in wavelength (e.g. positively to dominant wavelengths above 600 nm (in typical humans, termed 'red'), but negatively to wavelengths below 560 nm ("yellow-green"), or of reversed sign. Conversely, when defined strictly, such color selective neurons do not respond selectively to independent variations in luminance (the perceptual correlate of light intensity, when corrected for the sensitivity of the eye itself). Many neurons will respond to variations in both luminance and wavelength, but to varying extents. Nevertheless, to help distinguish whether a given neural response responds to variations in wavelength versus luminance, it has been standard to use flicker photometry in each subject to equate the luminance between the wavelength-varying stimuli, and to test for null responses at that "equal luminance" value (a luminance varying response), or to variations in wavelength across a range of luminance values spanning the equal luminance value. These two dimensions (wavelength and luminance) comprise two of the three independent axes of color space; spectral purity (psychophysical "saturation") comprises the third axis.

It is not clear that the authors understand these fundamental concerns and related experimental issues. For instance, in lines 500-501, the authors state: "Six equiluminous stimulus sets were used: red, green, blue, yellow, black, and white squares." By definition, variations in black and white cannot be "equiluminous"; the black and white are variations in luminance, and it is not clear that the wavelength and purity referred to as color were equal in luminance.

For instance, this manuscript does not state how "equiluminous" values were measured. If equiluminous values are measured in humans, those values do not translate exactly to those in macaque, since the macaque have a different ratio and distribution of long- to mid-wavelength sensitive cones compared to ~ 90% of humans, and the spectral sensitivity in macaques is correspondingly more protan relative to that in humans. Also, the various stimuli are described in terms of a select set of human experimenters, but those color percepts are likely different for macaques, due to these photoreceptor differences, plus differences in lens yellowing, etc. Thus the 'white' stimulus (as defined by humans) was likely 'reddish white' for the macaque.

Also, the "midspectral" colors (green and yellow) were less spectrally pure than the "red" and 'blue' (i.e. those 'colors' have a wider spectral range; and are closer to 'white' than the end-spectral colors, i.e. lower in saturation). Variations in spectral purity (or color contrast) can have a very strong influence on psychophysiological and physiological responses in 'color selective neurons.

Reply: Thank you for your meticulous explanation and suggestions. We apologized for the mistake in lines 500-501 (black and white stimuli should not be defined as ‘equiluminous’) and we have corrected the statement (Lines 627, Page 25). We have also provided a detailed description about how the luminance of the color stimuli in both fMRI and neurophysiological experiments were defined (equiluminous to the gray background) in the next paragraph (also in the manuscript, lines 547-560, on pages 22-23; lines 632-641 on pages 25-26). We also provided a direct comparison of the laminar results between the two stimulus sets: with equal physical luminance and with the luminance that minimizing the neural response from layers 4C α and 4B, in the third and fourth paragraph (also in the manuscript, lines 337-349 on page 15).

In the fMRI and physiological experiments, the four colored stimuli (red, green, blue, yellow) were all physically equiluminous to the gray background (~ 10.6 cd/m² in the electrophysiology experiment, ~ 5.2 cd/m² in the fMRI experiment). To generate equiluminous hue stimuli and take monitor gamut into consideration, we used a spectroradiometer (PR-670 SpectraScan, Photo Research Inc.) to ensure all colors were equiluminous with chosen RGB ratio and measured the real parameters of CIE1931 color space. We further calculated the cone excitations and cone contrasts of all stimuli based on CIE coordinates and luminance (Cole & Hine, 1992). As a summary, we now have provided the parameters of stimulus in multiple color spaces in Table R3.1 (also in Supplementary Table 2) and showed the cone excitations in the MacLeod-Boynton chromaticity diagram in Figure R3.1 (also in Supplementary Fig. 9).

Table R3.1 Parameters of stimulus in multiple color spaces in electrophysiology and fMRI experiments

	Electrophysiology				fMRI			
	Hue(°)	RGB	CIE1931 (x, y, L)	Cone Contrast (L,M,S)	Hue(°)	RGB	CIE1931 (x, y, L)	Cone Contrast (L,M,S)
Red	0	(160.7, 0, 0)	(0.61, 0.35, 10.99)	(0.32, -0.31, -0.91)	0	(176, 0, 0,)	(0.65, 0.34, 5.20)	(0.27, -0.47, -0.95)
Green	120	(0, 45, 0)	(0.29, 0.59, 10.44)	(-0.05, 0.35, -0.84)	120	(0, 130, 0)	(0.38, 0.57, 5.17)	(-0.03, 0.20, -0.89)
Blue	240	(0, 0, 255)	(0.15, 0.07, 10.02)	(-0.24, -1.80, 6.90)	240	(0, 0, 255)	(0.14, 0.09, 5.15)	(-0.23, -1.20, 9.83)
Yellow	60	(36.5, 36.5, 0)	(0.39, 0.52, 10.60)	(0.03, 0.22, -0.86)	60	(117.3, 117.3, 0)	(0.45, 0.49, 5.22)	(0.03, 0.07, -8.44)
Black	-	(0, 0, 0)	(0.24, 0.19, 0.08)	(-0.99, -1.00, -0.98)	-	(0, 0, 0)	(0.38, 0.33, 0)	(-1, -1, -1)
White	-	(61.2, 61.2, 61.2)	(0.28, 0.31, 20.40)	(0.93, 0.92, 0.96)	-	(137.7, 137.7, 137.7)	(0.35, 0.34, 10.1)	(0.96, 0.79, 1.36)
Gray	-	(30.6, 30.6, 30.6)	(0.28, 0.31, 10.55)	(0, 0, 0)	-	(117.3, 117.3, 117.3)	(0.34, 0.37, 5.22)	(0, 0, 0)

Figure R3.1 Stimuli of equal luminance in Macleod-Boynton color space

As your nice suggestion of the flicker photometry method in humans, we defined another set of stimuli based on the neural response from the M pathway (i.e. L4C α & L4B in V1) in the luminance-matching experiment to roughly equate the subjective luminance between the wavelength-varying stimuli. In the luminance-matching experiment, we used fast square-wave drifting grating (SF = 2Hz, temporal frequency = 16.67Hz, preferred orientation) composed of background gray (10.6cd/m²) and red/green/blue/yellow with multiple levels of luminance. Based on the responses from the M pathway, for each color, we chose the luminance value that evoked the minimal response from L4C α & L4B (vertical dashed black line in Figure R3.2) as the estimated “subjectively equal luminance”. This procedure mimics the flicker photometry method in humans. Then we measured the V1 laminar response activated by uniform squares with color luminance defined as “subjectively equal luminance”. In this way, we minimized the contamination of the response from luminance-selective cells in the M pathway. And we showed the laminar results of the luminance matched by the M pathway together with the laminar results of physically equal luminance in Figure R3.3 (also in Supplementary Fig.6) to make a direct comparison.

Figure R3.2 Response from V1 layer 4C α and 4B in the luminance-matching experiment.

As a result of the control experiment with stimulus luminance matched by response in M pathway, the laminar distributions of end-spectral bias and polarity bias were very similar to those activated by physically equiluminous stimuli: red-green bias and blue-yellow bias were strongest in L4C β (Figure R3.3, also in Supplementary Fig.7). The consistent results further confirmed the pre-cortical source and dominant P-pathway contribution of end-spectral bias. Due to the common use of physically equiluminous color in previous studies (Valverde Salzmann et al. 2012; Tootell et al. 1988; Liu et al. 2020) and in our fMRI experiment, we kept the results of physically equiluminous color (Figure R3.3) in the main text of the manuscript and moved the results of M-pathway-equiluminous colors to the supplementary materials (Supplementary Fig. 6-7). It is hard for us to perform the corresponding control experiment of subjective luminance in fMRI because we cannot separate the response of the M pathway from the BOLD signals. We have added the limitation to the discussion (line 464, page 19).

Figure R3.3 Laminar pattern of end-spectral bias and black dominance by using two sets of stimuli.

a. Laminar distribution of bias indices with stimulus luminance matched by M pathway. **b.** Laminar distribution of bias indices of with physically equiluminous stimulus (same as Figure 5 in the manuscript).

Finally, the dominant wavelengths of the four colors are shown in the CIE space (Figure R3.4, also in Supplementary Fig.9), and the spectral purity of colors is: red = 0.95, blue = 0.94, green = 0.84, and yellow = 0.88. Although the rank ordering of purity is similar to the ordering of response strength in V1 (Figure 1), the minor difference (0.01) between red and blue seems unlikely to evoke the significant difference in neural response in MUA. Based on the previous study (Tootell et al. 1988), the end-spectral bias still exists after controlling the purity. Therefore, the purity is not the reason for the end-spectral bias. Furthermore, when comparing the end-spectral bias in different brain regions, the findings about the bias transition across visual regions are not influenced by the purity due to the same purity value for each color pair. The effect of color purity is an interesting point, therefore, we have added the corresponding explanation in the Discussion (line 376-386, page 16).

Figure R3.4 Stimuli in the CIE xyY space.

Hollow circles represent isoluminant stimuli in CIE 1931 space used in the electrophysiological experiment. Solid circles represent the dominant wavelengths corresponding to each hue (blue: 466 nm, green: 549 nm, yellow: 567 nm, red: 605 nm).

2. Throughout the manuscript, the authors imply or state that the LGN is a beginning stage in the processing of color-opponency and/or the end-spectral bias. For instance, in the abstract, the authors state: “Our results suggest that the end-spectral bias . . . emerges in the LGN, and then, such bias is transmitted into V1. .”. Later, the authors state: ““The stronger spiking activity produced by red/blue than green/yellow was first found in the macaque primate visual cortex (Yoshioka & Dow, 1996)”. Later: “. .as previous studies suggest, V1 is the first place where mixing between the cone-opponent mechanism occurs (Lennie et al., 1990)”.

It would be helpful if the authors could more explicitly acknowledge that color opponency arises at least as early as in the retina (in horizontal cells; see MacNichol and Svatichin, 1956) and in ganglion cells in macaque (Gouras, others). It is quite possible (perhaps likely) that the end-spectral bias arises from simple subtractive combinations of the three different photopigments in retina –though this has not been widely tested yet. In contrast, neurons in the LGN have been regarded as “relay” cells, from retina to cortex. This issue could be easily resolved by re-wording some parts of the manuscript.

Reply: Thank you for your thoughtful suggestions. First, we now have explicitly acknowledged that color opponency arises at least as early as in the retina in the Introduction (in lines 55-57 on page 3). We removed the word ‘first’ from the sentence ‘The stronger spiking activity produced by red/blue than green/yellow was found in the macaque primate visual cortex’ in lines 366 on page 16.

Secondly, we admitted that in the current study, we can only conclude that end-spectral bias already exists in LGN rather than emerges from LGN. We have rephrased the sentence (line 40 on page 2) in the abstract by replacing the word ‘emerges’ with ‘exists’. And we also added the possibility that the end-spectral bias arises from simple subtractive combinations of the three different photopigments in the retina in the Discussion (lines 425-427 on page 18).

3. Beyond V1, the authors model and sample activity from “V2” and higher stage areas, without taking into account that color vision processing (and the end- versus mid-spectral sensitivity differences) are (to some extent) segregated into patches which occupy only a subset of each of these areas, and have different connections with other areas), especially in the stripes in V2. Nasr and Tootell (2018, Journal of Neuroscience) discuss and measure this end- versus mid-spectral bias in human fMRI. Such color-selectivity in V2 thin stripes is also shown earlier in awake behaving macaque (Tootell et al 2004, Cerebral Cortex). Another report (Tootell et al 1988) quantifies the end-versus mid-spectral differences in different layers in macaque V1, and shows the color bias in each of the six layers of LGN (Figure 9).

Reply: We thank the reviewer for pointing this issue out. As mentioned by the reviewer, V2 contains stripes (and V4 shows globs), which may exhibit differences in color bias. However, due to the limitations of the spatial resolution in our study (1.5mm isotropic), it was challenging to reveal and investigate these stripes [reported to be 1-1.5mm in macaques (Roe and Ts'o 1997)]. Note that Conway, Moeller, and Tsao (2007) identified stripes in V2 in macaques using a higher spatial resolution of 1.25 mm isotropic. Furthermore, it was not feasible for us to achieve a higher spatial resolution below 1.0 mm due to technical constraints. Moreover, even at 1.25 mm resolution, with the current equipment,

we would not be able to cover all the regions of interest adequately and achieve a reasonable Signal-to-Noise Ratio (SNR) [only approximately 57.87% of the current setting due to the small voxel size: $(1.25 \times 1.25 \times 1.25\text{mm}) / (1.5 \times 1.5 \times 1.5\text{mm})$].

In our attempt to investigate the stripes with the available data, we encountered additional challenges due to the nature of our experimental design. Unlike previous studies that employed block designs with grating stimuli, we conducted an event-related design using uniform square stimuli, which complicated the definition of stripes. Nonetheless, to investigate potential differences among subregions, we categorized the voxels within each ROI based on the difference between the mean responses to the four colors and the mean response to black and white stimuli. Our findings indicated that, except for the red-green bias in V2, there were no significant differences in bias indices between the two groups of voxels (Figure R3.5).

To further explore potential differences among sub-regions in V2 and V4, we analyzed another set of unpublished data utilizing grating stimuli [Figure R3.6, red/gray grating (equiluminant) vs 75% achromatic grating (equal mean luminance to red/gray grating), 1.5mm isotropic]. By using red/gray grating versus 75% achromatic grating as the contrast, we did not observe clear patterns corresponding to the stripes in V2, as reported in previous studies (Conway, Moeller, and Tsao 2007). Similar to our previous analysis, we divided the voxels within each ROI into high- and low-color selective groups based on their color selectivity. Then, we compared the bias indices between these two groups. The results revealed differential red-green bias, blue-yellow bias, and black-white bias in V2, blue-yellow bias and black-white bias in V1, as well as black-white bias in V4 between the high- and low-color selective groups (Figure R3.7).

Based on our current methodologies and supplementary analyses, we found differences in bias indices between the high- and low-color selective voxels in the ventral visual pathway, especially V2. Note that both groups of voxels exhibited end-spectral bias. Previous studies have reported that thin stripes in V2 showed stronger responses to end-spectral hues, while interstripes prefer mid-spectral hues (Yoshioka and Dow, 1996). The difference between our findings and previous studies may be attributed to spatial resolution limitations, impeding the discernment of finer subdivisions within V2. Consequently, our results may be predominantly influenced by thin stripes. Moreover, the use of uniform square stimuli in our experiment, different from the drifting gratings employed in previous investigations, may also contribute to a predominance of response patterns associated with thin stripes since both thick stripes and interstripes are sensitive to moving stimuli. To further investigate and compare end-spectral bias across sub-regions within V2 (and V4), future studies may employ higher spatial resolution and specific methodologies.

Tootell's work (2004, 2018) with technology advances is highly valuable and instructive. Furthermore, Tootell's work (1988) in LGN and V1 is also pioneering for the subcortical source of end-spectral bias. We have cited the reference in the Introduction (line 75-77, page 4) as a strong evidence of the feedforward hypothesis for end-spectral bias. The highly segregated horizontal patches in V1 to V4 together with the V1 laminar structure for color processing also suggest the possible laminar difference between patch and interpatch region in V2 and V4, which exacerbates the future need of multiple-technology combinations. Neurophysiological laminar recordings in V1, V2, and V4 guided by imaging results of stripe/patch locations are indispensable to uncover the possible dissociated pathway of color representation between the patch and interpatch region in V2 and V4. We have added the limitation of current work, technology advantage of Tootell's work and the further directions in the discussion (lines 462-474, page 19).

Figure R3.5 The comparison of bias indices for low and high color selective voxels.

a-c. The bias indices for low and high color selective voxels in each ROI divided according to the difference between the mean of four colors and the mean of black and white stimuli in the current data. ns, not significant, * $p < 0.05$ in comparison of GLMM analysis. Error bars indicate standard error.

Figure R3.6 No clear patterns corresponding to the stripes observed in V2.

V2 ROIs are encircled by white lines for two stimulus configurations (left column: the square aligned to the left horizontal meridian, right column: the square aligned to the right horizontal meridian) in monkey Q and P.

Figure R3.7 The comparison of bias indices for low and high color selective voxels.

a-c. The bias indices for low and high color selective voxels in each ROI divided according to red/gray grating versus 75% achromatic grating in another unpublished data. ns, not significant, * $p < 0.05$, ** $p < 0.01$, *** $p < 0.001$ in comparison of GLMM analysis. Error bars indicate standard error.

4. Part of the analysis (lines 535-540) is as follows: “For each stimulus configuration, in which there were two squares, we defined two sets of sub-ROIs (one on the left hemisphere and another on the right hemisphere, corresponding to two small squares). That is, in total, there were four sub-ROIs for

each color in one brain area (e.g., V1), corresponding to the four locations of squares. For each sub-ROI, based on the averaged responses across all six colors, the top 20 visual responsive voxels were selected...”

In the relevant Figure 3C, I also see four (sub) regions of interest. But given the known retinotopic organization of macaque cortex, relative to the configuration of the fixation point and the two stimuli below it (e.g. Figure 1, 2 etc.), I interpret only the two dorsalmost patches (along the V1-V2 border) of fMRI activity as driven by the stimuli of interest. The ventralmost two patches are instead located exactly at the confluent representation of the fovea, shared by V1, V2, V3. And V4, thus due to complicated visual effects on and near the foveal fixation point in the stimulus (as in most fMRI studies using a fixation task). Thus the ventralmost two of those sub-ROIs should be excluded from subsequent analysis.

Reply: We are grateful to the reviewer for this valuable comment. The stimuli used in our experiments were presented within the 2-6 degrees range. As the reviewer pointed out, due to the retinotopy in V4, the defined ROIs in V4 (specifically, two ventralmost patches) might encompass a portion of the fovea region. To address this possibility, we compared these two ROIs with the retinotopic probability distribution of V4 reported in the literature (Janssens et al. 2014). The analysis revealed a substantial overlap: most voxels were predominantly localized within the V4 region with high retinotopic probability (see Figure R3.8 for an example from Monkey Q). These results suggested that the involvement of the fovea region should be limited. Additionally, we observed distinct responses to different stimuli in these two ROIs, which also showed some differences from those in V1 and V2. Note that the fixation point was consistently presented throughout the experiment. Subjects maintained fixation well and our data were contrasted with the baseline. Therefore, the findings in two ROIs were unlikely to be primarily attributed to the fovea contributions. Furthermore, we did not observe any discernible distribution trend from the center to the periphery within the ROI, as discussed in our response to Reviewer #1 Q#10 (Figure R1.7).

Taken together, the two ventralmost patches were primarily contributed by V4, with limited contributions from the fovea.

Figure R3.8 Comparison of V4 ROIs with the V4 retinotopic probability map from the previous study (Janssens et al., 2013) in macaque Q.

The black star marks the approximate location of the fovea. The borders of the V4 ROI of monkey Q are indicated by the black dotted lines.

Minor Concerns:

1. As I understand it, the group infers the laminar source of their signals, without direct localization such as electrical lesions or other markers of exact microelectrode tip location. Although in some cases such inferences are likely. However, tissue dimpling from the electrode array and the inherent ambiguity of localization of the extracellular sources can make such inferences less certain.

Reply: Thank you for pointing out the concern. We calculated relative depth for each recording session and assigned cortical layers according to the dynamic signatures of MUA and CSD driven by rapidly flashed grating patches and small light spots, which has been found to be a solid basis (Wang et al., 2020). The method for laminar assignment has taken care of the different thicknesses that is due to tissue dimpling, stretching, or different probe placements. Based on the method (Wang et al. 2020), the laminar response patterns for awake monkey V1 is highly consistent with those for anaesthetized monkeys measured by single electrodes with anatomical reconstructions for cortical layers (Ringach,

Shapley, and Hawken 2002; Xing et al. 2012; Xing, Yeh, and Shapley 2009). Furthermore, according to the signatures of neural responses to stimuli at different orientations (Wang et al., 2020), L4Ca and L4Cb exhibit distinct properties which also justifies the method we used for laminar assignment in awake V1.

2. I do not understand this sentence; perhaps it can be re-worded: “Rather than equally encoding color information, the primary visual cortex (V1) exhibits strong bias on end-spectral colors (red and blue) relative to mid-spectral colors (green and yellow), which is similar to the black dominance found in the luminance domain.”

Reply: Thank you. To avoid any confusion and to reduce the unnecessary explanation for the unimportant concept, we have removed the “, which is similar to the black dominance found in the luminance domain” from the sentence (line 52-55, page3). And we have rewritten the sentence as: **“Rather than uniform representation of spectral information, the primary visual cortex (V1) exhibits asymmetric coding of different wavelengths of light: a strong bias on end-spectral colors (e.g., red and blue) relative to mid-spectral colors (green and yellow)”** (lines 52-54, page 3).

3. The interpretation of responses to “red-green” versus “blue-yellow” (e.g. Figure 2) was challenging, because two stimulus properties (instead of only at a time) co-vary; it is not obvious whether a given difference is due to a change in yellow or blue or green or red.

Reply: We are grateful for the reviewer's comments. To enable comparisons across regions, we presented the signals evoked by the four colors across regions in Figure R1.10 and Figure S4. As the reviewer pointed out, the observed changes in end-spectral bias from LGN to V4 were associated with the concurrent changes in responses to the paired two colors. For example, the responses to red and green exhibited a more convergent pattern from V1 to V4, accompanied by a decrease in responses to red and an increase in responses to green. We have added this information in the revised manuscript (lines 236-239, page 10).

4. The references for the projection of LGN (magno and parvo) to striate layers 4C (alpha and beta, respectively), e.g. lines 51-54, should include multiple primary studies by Jennifer Lund's group.

Reply: Thank you for your suggestions. We have cited the references for the projection from LGN to V1 L4C (Blasdel and Lund 1983; Lund, Angelucci, and Bressloff 2003) in both result and discussion section (line 306-307, page 13; line 411-412, page 17).

5. I don't understand the terminology used in the title and throughout this manuscript, referring to the end-spectral versus mid-spectral difference in wavelength effectiveness. Qualitatively, this effect is 'symmetrical' (on both sides of the midpoint), although there may be smaller quantitative variations in the extent of this symmetry on each side, which may also vary in different areas (is this what the authors are referring to?)

Reply: 'asymmetric coding' refers to the end-spectral bias. we did not intend to compare the strength of red bias and blue bias. We have added an explanation for 'asymmetric coding' in the Introduction as **"Rather than uniform representation of spectral information, the primary visual cortex (V1) exhibits asymmetric coding of different wavelengths of light: a strong bias on end-spectral colors (e.g., red and blue) relative to mid-spectral colors (green and yellow)"** in lines 52-54 on page 3.

6. The authors did not cite a 'first' description of the color bias (Kruger and Gouras, 1980, J Neurophysiol). Another relevant reference is Yoshioka, Dow and Vautin, 1996.

Reply: Thank you for your suggestions. We have cited the two references in lines 54-55, page 3; 366-367, page 16.

Reference

- Arcaro, M. J., and M. S. Livingstone. 2017. 'A hierarchical, retinotopic proto-organization of the primate visual system at birth', *Elife*, 6.
- Blasdel, G. G., and J. S. Lund. 1983. 'Termination of Afferent Axons in Macaque Striate Cortex', *Journal of Neuroscience*, 3: 1389-413.
- Cole, G. R., and T. Hine. 1992. 'Computation of Cone Contrasts for Color-Vision Research', *Behavior Research Methods Instruments & Computers*, 24: 22-27.
- Conway, B. R., S. Moeller, and D. Y. Tsao. 2007. 'Specialized color modules in macaque extrastriate cortex', *Neuron*, 56: 560-73.

- Girard, P., P. A. Salin, and J. Bullier. 1991. 'Visual activity in macaque area V4 depends on area 17 input', *Neuroreport*, 2: 81-4.
- Hayes, A. F., and K. J. Preacher. 2014. 'Statistical mediation analysis with a multicategorical independent variable', *Br J Math Stat Psychol*, 67: 451-70.
- Janssens, T., Q. Zhu, I. D. Popivanov, and W. Vanduffel. 2014. 'Probabilistic and single-subject retinotopic maps reveal the topographic organization of face patches in the macaque cortex', *J Neurosci*, 34: 10156-67.
- Jayakumar, J., B. Dreher, and T. R. Vidyasagar. 2013. 'Tracking blue cone signals in the primate brain', *Clin Exp Optom*, 96: 259-66.
- Kremkow, Jens, Jianzhong Jin, Stanley J. Komban, Yushi Wang, Reza Lashgari, Xiaobing Li, Michael Jansen, Qasim Zaidi, and Jose-Manuel Alonso. 2014. 'Neuronal nonlinearity explains greater visual spatial resolution for darks than lights', *Proceedings of the National Academy of Sciences*, 111: 3170-75.
- Kriegeskorte, N., W. K. Simmons, P. S. Bellgowan, and C. I. Baker. 2009. 'Circular analysis in systems neuroscience: the dangers of double dipping', *Nat Neurosci*, 12: 535-40.
- Leite, F. P., D. Tsao, W. Vanduffel, D. Fize, Y. Sasaki, L. L. Wald, A. M. Dale, K. K. Kwong, G. A. Orban, B. R. Rosen, R. B. Tootell, and J. B. Mandeville. 2002. 'Repeated fMRI using iron oxide contrast agent in awake, behaving macaques at 3 Tesla', *Neuroimage*, 16: 283-94.
- Liu, Y., M. Li, X. Zhang, Y. Lu, H. Gong, J. Yin, Z. Chen, L. Qian, Y. Yang, I. M. Andolina, S. Shipp, N. McLoughlin, S. Tang, and W. Wang. 2020. 'Hierarchical Representation for Chromatic Processing across Macaque V1, V2, and V4', *Neuron*, 108: 538-50 e5.
- Lund, J. S., A. Angelucci, and P. C. Bressloff. 2003. 'Anatomical substrates for functional columns in macaque monkey primary visual cortex', *Cereb Cortex*, 13: 15-24.
- Mullen, K. T., S. O. Dumoulin, and R. F. Hess. 2008. 'Color responses of the human lateral geniculate nucleus: selective amplification of S-cone signals between the lateral geniculate nucleus and primary visual cortex measured with high-field fMRI', *European Journal of Neuroscience*, 28: 1911-23.
- Ringach, D. L., R. M. Shapley, and M. J. Hawken. 2002. 'Orientation selectivity in macaque V1: diversity and laminar dependence', *J Neurosci*, 22: 5639-51.
- Roe, Anna Wang, and Daniel Y. Ts'o. 1997. 'The Functional Architecture of Area V2 in the Macaque Monkey.' in Kathleen S. Rockland, Jon H. Kaas and Alan Peters (eds.), *Extrastriate Cortex in Primates* (Springer US: Boston, MA).
- Solomon, S. G., and P. Lennie. 2005. 'Chromatic gain controls in visual cortical neurons', *J Neurosci*, 25: 4779-92.
- Tootell, R. B., M. S. Silverman, S. L. Hamilton, R. L. De Valois, and E. Switkes. 1988. 'Functional anatomy of macaque striate cortex. III. Color', *J Neurosci*, 8: 1569-93.
- Valverde Salzmann, M. F., A. Bartels, N. K. Logothetis, and A. Schuz. 2012. 'Color blobs in cortical areas V1 and V2 of the new world monkey *Callithrix jacchus*, revealed by non-differential optical imaging', *J Neurosci*, 32: 7881-94.
- Welvaert, M., and Y. Rosseel. 2013. 'On the definition of signal-to-noise ratio and contrast-to-noise ratio for FMRI data', *PLoS One*, 8: e77089.
- Xing, D., C. I. Yeh, S. Burns, and R. M. Shapley. 2012. 'Laminar analysis of visually evoked activity in the primary visual cortex', *Proc Natl Acad Sci U S A*, 109: 13871-6.
- Xing, D., C. I. Yeh, and R. M. Shapley. 2009. 'Spatial spread of the local field potential and its laminar variation in visual cortex', *J Neurosci*, 29: 11540-9.

Zeidman, P., A. Jafarian, N. Corbin, M. L. Seghier, A. Razi, C. J. Price, and K. J. Friston. 2019. 'A guide to group effective connectivity analysis, part 1: First level analysis with DCM for fMRI', *Neuroimage*, 200: 174-90.

REVIEWERS' COMMENTS

Reviewer #1 (Remarks to the Author):

1. My major concern previously had been that the color of the stimulus was very poorly described, including the display devices and calibration, and that there needed to be colorimetric definitions provided. The authors have now provided CIE specifications and display/calibration information (which appear appropriate – and the new experiments have used a variant of minimum motion very usefully provide a physiological as well as colorimetric calibration), as well as measures of cone activation, and the authors now more clearly note that because responses to the blue stimulus, for example, could be carried by 'red-green' opponent pathways, because the stimulus is a strong modulator of L- and M-cone activity as well as S-cones. I do think there could be better signposting still for readers, to explain the relationship between the subjective words used as stimulus descriptors ('blue', 'yellow', 'red', 'green') and the likely activity in cone-opponent retinal/subcortical pathways that provide input to cortex. For example even at line 99 where the authors note "black or white with the same absolute contrast", it would be useful to point in the text here (or in the legend to Figure 1) to Supp Table 2, for the L,M, and S cone contrast values of all the stimuli.

I am not sure what the units of the cone contrast in Supp Table 2 are (please make clearer), however they and Supp Fig 9 do indicate that while the black and white stimuli have roughly equal cone contrast, the total cone contrast for the red stimulus is substantially more than that for the green stimulus, and the cone contrast of the blue stimulus is substantially greater than that for the yellow stimulus. The authors could consider more clearly the fact that end spectral bias (larger response to red and blue stimuli than green and yellow) reflects the larger cone-contrasts for the stimuli used in the measurements. Greater homogeneity in higher visual cortical areas may, for example, reflect shifts away from cone-opponent coding.

2. A second concern had been that differences in color bias along the cortical hierarchy (which is most obvious when comparing MRI signals in V1 and V4) may reflect eccentricity-dependent changes in the properties of voxels used in analysis (ie. the voxels may be drawn from different regions of the visual field in different brain areas). In the author's response they make clear that experimental complexity means that they were unable to obtain retinotopic maps in the MRI scanner for these animals, and so are unable to know precisely which part of the visual field is represented in the voxels that are responsive to the colored stimuli. This is fair enough, and the authors provide rational arguments about why eccentricity may not explain the changes in bias between brain regions, but I think they should also be much clearer in the text that they do not have retinotopic maps, and therefore are unable to ensure that the same parts of the visual field are being assessed in each area.

The additional concern that arises from the lack of retinotopic maps is that while there is much reproducibility in visual field maps across individuals, there is also considerable variability, so without receptive field maps to establish visual field reversals, it is difficult to know precisely where functional areal boundaries lie. It needs to be made clearer in the main Results text that the areal boundaries are defined from average atlases, not the individual animals, and are therefore not precisely known in these

experiments. Additionally, during the fMRI measurements, one of the stimuli was near the vertical meridian, that is at the boundary of cortical areas, resolution of which responses belong to which area can be difficult. These limitations need to be made clear and the data presented more transparently – for example Figures 1 and 3 present V1 and V2/V4 data respectively, however it is difficult to understand these representations as far as I can tell they are only partial representations of the data (omitting V2/V4 or V1 respectively). I would prefer to see, for each of the 4 animals, the full, perhaps unthresholded t-map of these occipital areas during the presentation of stimuli (ie. in at least the supplementary data one map, not two separate maps, with the putative ROIs for V1, V2 and V4 indicated/overlaid). The authors could also rerun their analyses having shifted the putative areal boundaries by some value – do the results depend on the accuracy of the atlas alignments? In sum, what is the confidence that the authors have that the responses they analyse are in separate cortical areas, and how would changing their assumption about the location of areal boundaries change the results that they obtain.

Reviewer #2 (Remarks to the Author):

Comments regarding whether revisions adequately address previous concerns:

The thorough and detailed responses from authors is much appreciated. The diligence taken in addressing initial comments and concerns raised is evident in the many in-text additions, changes to existing figures, and the additional supplementary figures. The additional laminar recordings control experiment provides an insightful explanation for the noise differences present in the original intralaminar recordings between chromatic and achromatic trials. There are no further revisions requested.

Comments regarding revisions addressing Reviewer #3 concerns raised in initial review:

The main concerns raised by R3 have been addressed with the additional control experiment, in-text rephrasing, and substantial additions of noteworthy points made to the discussion text. Specific references have been included as requested, as well as other additional relevant references. An improved description of how the color stimuli luminance was matched and defined is now provided. The described procedure in the revised manuscript is adequate and in line with standard practices for vision science experiments focused on luminance and color comparisons. Authors have addressed spectral purity concerns, providing a coherent argument against any spectral purity discrepancies driving the end-spectral bias results, which is now included in the discussion. In summary, all main and minor concerns of R3 have been adequately addressed to the extent possible with the additional experiment, analyses, or with text revisions and additions to the discussion section.

Reviewer #1 (Remarks to the Author):

1. My major concern previously had been that the color of the stimulus was very poorly described, including the display devices and calibration, and that there needed to be colorimetric definitions provided. The authors have now provided CIE specifications and display/calibration information (which appear appropriate – and the new experiments have used a variant of minimum motion very usefully provide a physiological as well as colorimetric calibration), as well as measures of cone activation, and the authors now more clearly note that because responses to the blue stimulus, for example, could be carried by ‘red-green’ opponent pathways, because the stimulus is a strong modulator of L- and M-cone activity as well as S-cones. *I do think there could be better signposting still for readers, to explain the relationship between the subjective words used as stimulus descriptors (‘blue’, ‘yellow’, ‘red’, ‘green’) and the likely activity in cone-opponent retinal/subcortical pathways that provide input to cortex. For example even at line 99 where the authors note “black or white with the same absolute contrast”, it would be useful to point in the text here (or in the legend to Figure 1) to Supp Table 2, for the L,M, and S cone contrast values of all the stimuli. I am not sure what the units of the cone contrast in Supp Table 2 are (please make clearer)*, however they and Supp Fig 9 do indicate that while the black and white stimuli

have roughly equal cone contrast, the total cone contrast for the red stimulus is substantially more than that for the green stimulus, and the cone contrast of the blue stimulus is substantially greater than that for the yellow stimulus. *The authors could consider more clearly the fact that end spectral bias (larger response to red and blue stimuli than green and yellow) reflects the larger cone-contrasts for the stimuli used in the measurements. Greater homogeneity in higher visual cortical areas may, for example, reflect shifts away from cone-opponent coding.*

Reply: Thank you for your suggestions. We now added a clear reference to Supplementary Table 1 for cone contrast values of all the stimuli used in this study (please see changes lines 85-87 on page 5). The cone contrast for a particular cone class here was defined as the ratio of the cone excitation due to the square stimulus relative to that due to the background. Therefore, there is no unit for the cone contrast here. We have clarified this by adding the definition of cone contrast in the text (lines 512-513, page 21).

For your third suggestion, we agree that end-spectral bias reflects the larger cone contrasts for the stimuli used in the measurements because, as we showed in the previous response letter, red and blue stimuli have larger values on the L-M cone contrast. We have made a more precise statement to reflect the fact in lines 376-378 on page 16. The fact you pointed out helps us give a more straightforward explanation of end-spectral bias and homogeneity patterns along the visual hierarchy, which has been added in lines 452-453 on page 19.

2. A second concern had been that differences in color bias along the cortical hierarchy (which is most obvious when comparing MRI signals in V1 and V4) may reflect eccentricity-dependent changes in the properties of voxels used in analysis (ie. the voxels may be drawn from different regions of the visual field in different brain areas). In the author's response they make clear that experimental complexity means that they were unable to obtain retinotopic maps in the MRI scanner for these animals, and so are unable to know precisely which part of the visual field is represented in the voxels that are responsive to the colored stimuli. This is fair enough, and the authors provide rational arguments about why eccentricity may not explain the changes in bias between brain regions, but *I think they should also be much clearer in the text that they do not have retinotopic maps, and therefore are unable to ensure that the same parts of the visual field are being assessed in each area.*

The additional concern that arises from the lack of retinotopic maps is that while there is much reproducibility in visual field maps across individuals, there is also considerable variability, so without receptive field maps to establish visual field reversals, it is difficult to know precisely where functional areal boundaries lie. *It needs be made clearer in the main Results text that the areal boundaries are defined from average atlases*, not the individual animals, and are therefore not precisely known in these experiments. *Additionally, during the fMRI measurements, one of the stimuli was near the vertical meridian, that is at the boundary of cortical areas, resolution of which responses belong to which area can be difficult. These limitations need to be made clear and the data presented more transparently* – for example Figures 1 and 3 present V1 and V2/V4 data respectively, however it is difficult to understand these representations as as far as I can tell they are only partial representations of the data (omitting V2/V4 or V1 respectively). *I would prefer to see, for each of the 4 animals, the full, perhaps unthresholded t-map of these occipital areas during the presentation of stimuli (ie. in at least the supplementary data one map, not two separate maps, with the putative ROIs for V1, V2 and V4 indicated/overlaid). The authors could also rerun their analyses having shifted the putative areal boundaries by some value* – do the results depend on the accuracy of the atlas alignments? In sum, what is the confidence that the authors have that the responses they analyse are in separate cortical areas, and how would changing their assumption about the location of areal boundaries change the results that they obtain.

Reply: We appreciate and agree with the valuable comments from the reviewer. To address the reviewer’s concerns and validate our results, we conducted the following revision and analyses:

We have made more explicit statements in the main results text (lines 212-213 on page 10) that we did not have retinotopic maps of each animal. we have added sentences to introduce the average atlases method to define areal boundaries. As the reviewer noticed, we aligned the individual dataset to the D99 atlas (an individual atlas) in NMT v2 space and then defined the functional areal boundaries based on this atlas. We agree with the review that the absence of receptive field maps made it challenging to precisely estimate the location of functional areal boundaries. As the reviewer suggested, we have explicitly clarified that the areal boundaries are derived from the atlas rather than individual animals in lines 211-214 on page 10, also as below.

“In this study, individual images were aligned to the symmetric NIMH Macaque Template (NMT) v2. The areal boundaries were established based on the D99 atlas in NMT v2 space rather than individual receptive field maps. Consequently, the precise accuracy of the areal boundaries might be subject to variation.”

As the reviewer mentioned, one of the stimuli was presented close to the vertical meridian, representing the boundary of cortical areas. As suggested by the reviewer, we have explicitly clarified that disentangling the area to which the responses correspond was a challenge due to the proximity to this boundary in the revised manuscript (lines 213-215 on page 10). Moreover, as the reviewer suggested, we presented the unthresholded t-map for each animal (Figure R1a below, also see Supplementary Fig. 2a in supplementary materials). It should be noted that, in the main results, voxels of ROIs were selected based on responses to presented stimuli (see the section of “Definition of regions-of-interest (ROIs)” in Method). To offer a comprehensive overview of our findings, we have also presented unthresholded percent signal change maps in the supplementary materials (Figure R1b below, also see Supplementary Fig. 2b in supplementary materials). The distribution of unthresholded t-maps and percent signal maps exhibited high similarities, with higher t-values corresponding to greater percent signal changes, albeit with subtle differences. Here, we noticed a mistake in Figures 1e, 2a, and 3a: the color presents the percent signal change instead of the t value. We have corrected this mistake in the revision.

Furthermore, we employed two additional analyses to evaluate the potential impact of the abovementioned limitations as well as the accuracy of atlas alignment, as pointed out by the reviewer, on our results.

Firstly, we aligned our data with another dataset shared by Janssens, Zhu et al. (2014) to shift the putative areal boundaries. Note that this dataset provides the probabilistic map of V1, V2, V3, and V4 based on the retinotopic maps from multiple monkeys. As shown in Figure R2 (also see Supplementary Fig. 6), the ROIs defined in the main results showed a high probability of representing V1, V2, and V4. To avoid potential impacts of alignment accuracy (particularly around the boundaries) and individual differences, we selected voxels demonstrating at least 75% probability within the defined ROIs in the main text to create a new set of ROIs with higher precise representations of V1/V2/V4. Subsequently, we performed the same analyses in the main text, yielding similar findings to the main

results (Figure 3). Please refer to Figure R3 (Supplementary Fig. 7) for details.

Secondly, to shift the putative areal boundaries, instead of using the V1/V2/V4 masks from the D99 atlas, we selected voxels demonstrating at least 75% probability to create 75% probability V1/V2/V4 masks based on the probabilistic map. Note that the new set of masks may exhibit V1/V2/V4 more precisely. Next, we defined ROIs within these more precise masks following the same methods as in the main text. Again, consistent results were found with those shown in the main Results (please see results in Figure R4 as well as in Supplementary Fig. 8).

Based on the above two additional analyses, while acknowledging that precise functional areal boundaries were not delineated in our study, as explicitly stated in the revised main results text, our results are expected to be less affected by this limitation. We have added the control results in the main text (lines 216-220 on page 10) and supplementary materials (Supplementary Notes, Supplementary Fig. 6-8).

Figure R1. Unthresholded t-map (a) and percent signal map (b) for each animal.

Note that only positive activations evoked by stimuli are shown. The borders of ROIs are encircled by white lines.

Figure R2. The probabilistic map of V1, V2, V3, and V4 modified from Janssens et al. (2014).

Note that voxels around the boundary are assigned to a visual region with the highest probability based on their probability in each region. The warm and cold colors represent V1/V3, and V2/V4, respectively, with the intensity of the color indicating the probability of the voxel representing the respective area. The borders of ROIs are encircled by white lines.

Figure R3. End-spectral and polarity bias in V1, V2, and V4 based on voxels demonstrating at least 75% probability within the defined ROIs in the main text.

a-c. The averaged fMRI responses to three color pairs across all four subjects in V1, V2, and V4, respectively. *** $p < 0.001$, GLMM analysis. **d.** The color homogeneity across V1, V2, and V4. * $p < 0.05$, *** $p < 0.001$, post hoc comparisons of GLMM with Bonferroni corrections. **e-g.** The change trends of red–green bias, blue–yellow bias, and black–white bias from V1 to V4. * $p < 0.05$, ** $p < 0.01$, post hoc comparisons of GLMM with Bonferroni corrections. Data are presented as mean \pm SE ($n = 102$ runs).

Figure R4. End-spectral and polarity bias in V1, V2, and V4 based on the defined ROIs based on 75% probability V1/V2/V4 masks.

a-c. The averaged fMRI responses to three color pairs across all four subjects in V1, V2, and V4, respectively. *** $p < 0.001$, GLMM analysis. **d.** The color homogeneity across V1, V2, and V4. *** $p < 0.001$, post hoc comparisons of GLMM with Bonferroni corrections. **e-g.** The change trends of red–green bias, blue–yellow bias, and black–white bias from V1 to V4. * $p < 0.05$, ** $p < 0.01$, post hoc comparisons of GLMM with Bonferroni corrections. Data are presented as mean \pm SE ($n = 102$ runs).

Reviewer #2 (Remarks to the Author):

Comments regarding whether revisions adequately address previous concerns:

The thorough and detailed responses from authors is much appreciated. The diligence taken in addressing initial comments and concerns raised is evident in the many in-text additions, changes to existing figures, and the additional supplementary figures. The additional laminar recordings control experiment provides an insightful explanation for the noise differences present in the original intra-laminar recordings between chromatic and achromatic trials. There are no further revisions requested.

Reply: We appreciate all of your comments.

Comments regarding revisions addressing Reviewer #3 concerns raised in initial review:

The main concerns raised by R3 have been addressed with the additional control experiment, in-text rephrasing, and substantial additions of noteworthy points made to the discussion text. Specific references have been included as requested, as well as other additional relevant references. An improved description of how the color stimuli luminance was matched and defined is now provided. The described procedure in the revised manuscript is adequate and in line with standard practices for vision science experiments focused on luminance and color comparisons. Authors have addressed spectral purity concerns, providing a coherent argument against any spectral purity discrepancies driving the end-spectral bias results, which is now included in the discussion. In summary, all main and minor concerns of R3 have been adequately addressed to the extent possible with the additional experiment, analyses, or with text revisions and additions to the discussion section.

Reply: Thank you very much for your comments.

References:

Janssens, T., Q. Zhu, I. D. Popivanov and W. Vanduffel (2014). "Probabilistic and single-subject retinotopic maps reveal the topographic organization of face patches in the macaque cortex." J Neurosci **34**(31): 10156-10167.